# Direct retrieval of Zernike-based pupil functions using integrated diffractive deep neural networks

Elena Goi ®[1,2] ✉, Steffen Schoenhardt ®[1,2] & Min Gu ®[1,2] ✉

Retrieving the pupil phase of a beam path is a central problem for optical systems across scales, from telescopes, where the phase information allows for aberration correction, to the imaging of near-transparent biological samples in phase contrast microscopy. Current phase retrieval schemes rely on complex digital algorithms that process data acquired from precise wavefront sensors, reconstructing the optical phase information at great expense of computational resources. Here, we present a compact optical-electronic module based on multi-layered diffractive neural networks printed on imaging sensors, capable of directly retrieving Zernike-based pupil phase distributions from an incident point spread function. We demonstrate this concept numerically and experimentally, showing the direct pupil phase retrieval of superpositions of the first 14 Zernike polynomials. The integrability of the diffractive elements with CMOS sensors shows the potential for the direct extraction of the pupil phase information from a detector module without additional digital post-processing.

Retrieving the phase distribution of an incoming wavefront is a problem of central interest for imaging systems across scales. On one hand, this phase information can be used in the context of phase-contrast imaging, the characterization of near-transparent specimen in biology and medical research[1], which has led to significant advances in live cell monitoring[2] and tissue imaging[3]. On the other hand, unwanted distortions of the wavefront result in the limited performance of imaging systems of any scale from microscopes to telescopes[4,5], which can be corrected for if known. The phase distribution across a wavefront can thereby be conveniently described by Zernike polynomials[6], which were first introduced by Frits Zernike (1953 Nobel Prize in Physics) in 1934 to describe the diffracted wavefronts in phase contrast microscopy. The Zernike polynomials form a complete basis set of functions that are orthogonal over a circle of unit radius, and therefore their linear combination offer a mathematical description of arbitrary pupil phase distributions of an optical system while yielding minimum variance over a circular pupil. Hence, the Zernike polynomials have been used in a wide range of field such as describing atmospheric

turbulence in astronomy[7,8], ophthalmic optics[9], lens design[10] and microscopy[11].

As phase retrieval is such a central problem to optical imaging systems, numerous methods for solving this task have been developed. Artificial Neural Networks (ANNs) and deep learning[12–14], with their capability to learn complex relationships without being programmed with specific physical rules, have been applied to determine the Zernike coefficients that represent a given wavefront since the '90 s[15]. Together with conventional phase retrieval algorithms, such as the Gerchberg-Saxton algorithm[16] and indirect optimization methods, such as modal wavefront sensing[17] or pupil segmentation[18], computer-based ANNs are used for direct[11,19] and indirect[20–22] pupil phase retrieval. In indirect phase retrieval, focal plane images or detected wavefronts[23] are mapped to Zernike coefficients in a modal-based approach, requiring a subsequent reconstruction of the wavefront. While indirect phase retrieval is the most common method to determine the phase of a wavefront, direct phase retrieval, where the wavefront phase information is directly retrieved, was recently shown

[1]Institute of Photonic Chips, University of Shanghai for Science and Technology, Shanghai 200093, China. [2]Centre for Artificial-Intelligence Nanophotonics, School of Optical-Electrical and Computer Engineering, University of Shanghai for Science and Technology, Shanghai 200093, China. ✉e-mail: elenagoi@usst.edu.cn; gumin@usst.edu.cn

to achieve higher accuracy[11,19] in a single forward propagation step. This higher accuracy does, however, come at the cost of computational complexity, large memory footprint[11,19] and the need to use a dual CCD camera system to reconstruct the sign of axially symmetric aberrations such as defocus, which requires modification to the beam path of the imaging system they are applied to[11,19].

Despite the computational power and the flexibility of electronic processing units and artificial intelligence (AI) accelerators, photonic hardware and optical neural networks[24–32] have been proposed as a paradigm shift for deep learning and, in general, machine learning, using photons instead of electrons for computation. Photonic hardware offers high-speed optical communication (speed of light in medium) and massive parallelism of optical signals (multiplexing in time, space, wavelength, polarization, orbital angular momentum, etc.). Through advances in photonic integration technology with low losses[33], this results in compact and configurable integrated photonic processors. Free space optical neural networks such as diffractive neural networks (DN₂s)[25,34,35] and convolutional neural networks[28], on the other hand, exploit the nature of light propagation in free space and the interaction of a light field with thin scattering layers to implement convolutions and matrix multiplications passively in the optical domain[33]. Among these, the more compact device form factors can be achieved by DN₂s, where the thin diffractive layers are separated only by a few wavelengths, able to implement all-optical machine learning tasks such as classification algorithms[25,36,37] or decryption[34] with record-high neural densities. These systems can passively process optical information in its native domain, with the advantages of direct information processing at the speed of light, without the need for image digitalization with specialized detectors or digital post-processing. However, while DN₂s commonly rely on phase modulation for information processing, the inputs considered in current experimental implementations[25,35,36,38] are intensity distributions, omitting the phase of the incident fields. Although there are several

proposals describing optical neural networks processing complex-valued inputs in recent literature, these works implement the neural networks either in-silico[11,39,40], as digital neural networks, or they are restricted to numerical models of D₂N₂s. The numerically proposed optical networks are focused on performing logical operations on vortex beams[41] or orbital angular momentum multiplexing schemes[42,43] and hence specialized on operations with radially symmetric wavefronts, rather than the retrieval of arbitrary wavefronts, which is the essential problem for imaging systems.

In this work, we propose a new single-step method to retrieve an arbitrary pupil phase of an optical beam path through the analysis of its Point Spread Function (PSF), performed by a DN₂ working in conjunction with an imaging sensor (Fig. 1a). From the complex-valued PSF received as input, the DN₂, which operates optically in the near-infrared (NIR) wavelength region, passively reconstructs an intensity distribution representing the corresponding pupil phase in sign and magnitude on the output. The resulting intensity distribution is then detected by the sensor, thus adding the nonlinear responsivity required to achieve an integrated diffractive deep neural network (ID₂N₂) module with phase retrieval capabilities. We show numerically and experimentally that these optoelectronic networks can retrieve the pupil phase of incoming PSFs with low error rates and, through co-integration with standard complementary metal oxide semiconductor (CMOS) imaging sensors, have the potential for leading to a new generation of compact optoelectronic wavefront sensors. Compared with previous works[11,23], our approach allows for direct pupil phase retrieval from a single compact optoelectronic sensing element.

## Results

The ID₂N₂ presented in this work is an optoelectronic neural network that combines a DN₂ composed by four planar diffractive elements with a nonlinear activation function (Fig. 1b). The DN₂ receives in input a complex field that was linearly transformed through an objective that

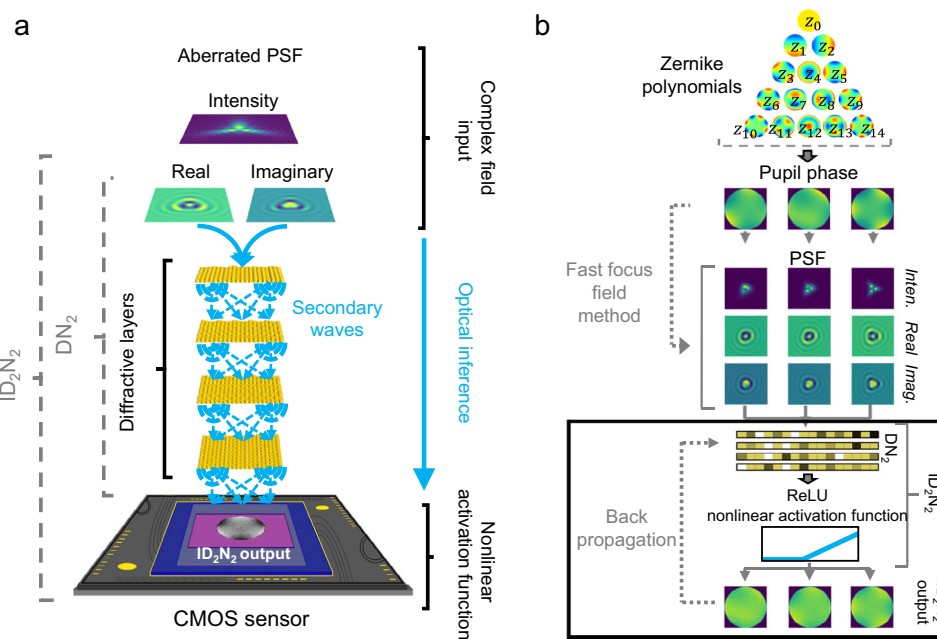

**Fig. 1 | Integrated Diffractive Deep Neural Networks (ID₂N₂). a** Schematic of a diffractive neural network integrated on a commercial CMOS chip performing pupil phase retrieval. An ID₂N₂ comprises a diffractive neural network (DN₂) and an activation function based on the nonlinear response of the CMOS detector module. The DN₂ consists of multiple diffractive layers, where each point on a given layer acts as a diffractive neuron, with a complex-valued transmission coefficient that can be trained by using deep learning to perform a function between the input and output planes of the network. **b** Flow diagram of ID₂N₂ training. The training dataset consists of a set of known pupil functions, built from linear combinations of Zernike polynomials from $Z_1$ to $Z_{14}$ (OSA/ANSI indices), and the corresponding point spread functions (PSF) calculated with a fast focus field method[44]. A mean square error loss function is defined to evaluate the performance of the ID₂N₂ with respect to the desired target. During the computer-based training, the complex transmission coefficient of each diffractive neuron of the ID₂N₂ is iteratively adjusted through an error backpropagation method.

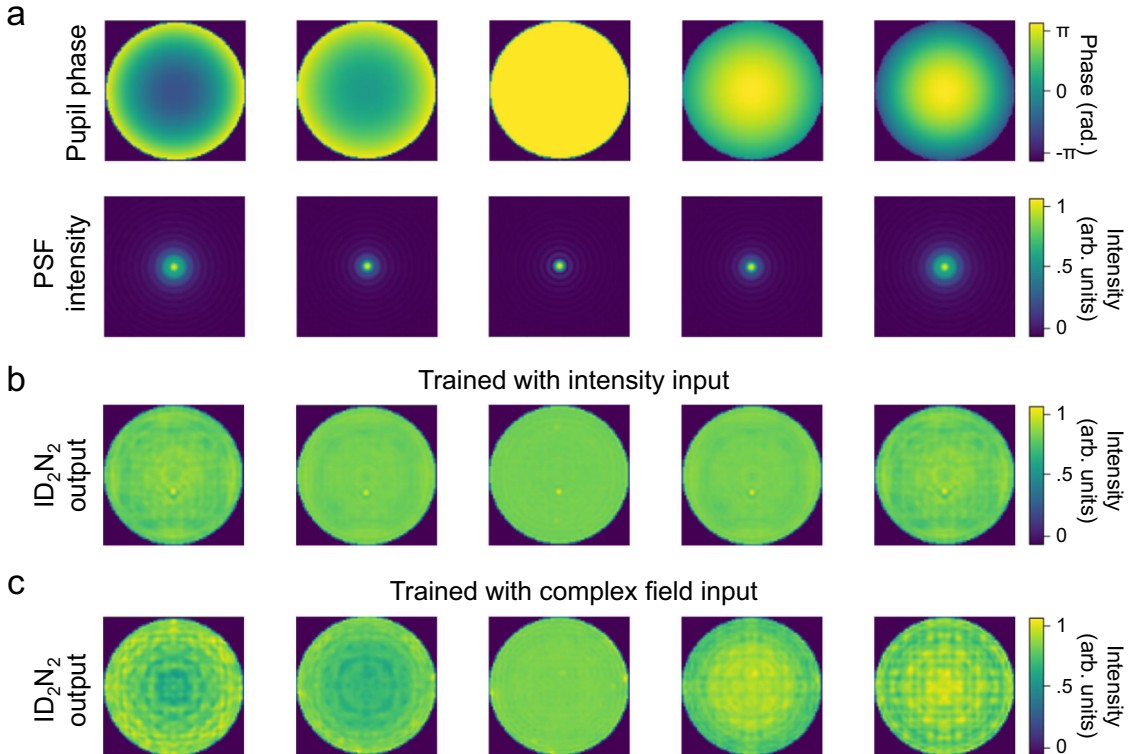

**Fig. 2 | ID$_2$N$_2$ non-degenerate response to defocus. a** Defocus (Z$_4$) terms applied and corresponding calculated PSF intensities. The amplitude is scanned between ±π radians, the PSF are calculated at the focal spot of a 10× 0.25NA objective for a point source of 100 μm diameter using a fast focus field method[44]. **b** Output intensities for the different defocus terms in (**a**) of an ID$_2$N$_2$ trained with PSFs intensity distributions dataset. **c** Output intensities for the different defocus terms in (**a**) of an ID$_2$N$_2$ trained with complex field PSFs dataset. Despite the fact that positive and negative defocus terms of the same amplitude give identical PSFs in intensity, the output of an ID$_2$N$_2$ trained with complex field PSFs is unambiguous. The simulations are performed at a wavelength of 785 nm.

forms a PSF, which is a linear expression of the pupil phase distortion in form of a single Zernike polynomial or a superposition of multiple Zernike polynomials. Retaining phase and amplitude information of the incoming light, the DN$_2$ can scatter and modulate each of a multitude of aberrated PSFs, mapping them into a specific output field. The intensity distribution of the output field is a linear representation of the original pupil phase in magnitude and sign. Through this optical inference process, the pupil phase is directly obtained in the sense that the phase distribution is shown immediately as a distribution over (x, y) on the output plane of the DN$_2$, rather than being represented as coefficients of Zernike polynomials, as done in indirect phase retrieval schemes.

The complex field PSFs used during training were calculated following a fast Fourier transform (FFT) implementation of vectorial Debye theory[44], while the complex field PSFs in the experiment were generated by an optical setup. While the nonlinear activation in the optical characterization of the printed DN$_2$s is implemented through the photoelectric conversion of the field incident on the CCD sensor[38], this behavior is approximated as a Rectified Linear Unit (ReLU) function of optimized shape (see Supplementary Fig. 1 for optimization) for purposes of in-silico training and numerical characterization of the DN$_2$s. The choice to use the ReLU function to approximate the nonlinearity of the detector is motivated by the response curve for a light-sensitive detector module consisting of photoelectric conversion and electronic readout circuit[45] (see Supplementary Information). While the diffractive neural network performs a linear operation on the light field, the nonlinear response of the CCD sensor at the output of the DN$_2$ may serve as nonlinear activation function of a hidden layer, if the readout of the sensor is used as input to another optical or digital neural network. The images obtained from the CCD sensor are a direct representation of the pupil phase in the optical characterization

systems and no further post-detection computation or post-processing is required to obtain the result.

## ID$_2$N$_2$ nondegenerate response to a varying degree of defocus

The ID$_2$N$_2$ are trained in silico and a detailed description of the forward model is given in Fig. 1b, Methods and Supplementary information sections. The numerical demonstration of the ability of the ID$_2$N$_2$ to reconstruct axially symmetric pupil phase distributions is reported in Fig. 2, where the pupil phase of the PSF and the outputs of the ID$_2$N$_2$ modules are shown for defocus terms (Z$_4$) of changing sign and amplitude. It is important to note that an ID$_2$N$_2$ trained using only the intensity pattern of the PSFs is not able to distinguish the sign of defocus, while the ID$_2$N$_2$ trained using the complex fields that describe the PSFs can retrieve the information contained within the phase structure of the PSF and encode it in the intensity of the outputs[23]. The outputs of ID$_2$N$_2$ trained using only the intensity patterns are smoother and more uniform compared to the outputs of a ID$_2$N$_2$ trained using the complex fields, since the network appears to map the ambiguous inputs in an image that is the average of the multiple outputs given during the training[46].

## ID$_2$N$_2$ numerical performance

After proving the ability of the ID$_2$N$_2$ trained with complex fields to retrieve the phase information of an incoming PSF by comparing the outputs when a symmetric pupil phase distributions with different signs, such as defocus, is applied at the focal spot, we work with PSFs calculated at 20 μm from the focal spot. Moving away from the focal spot, the PSFs increase in size and are therefore easier to characterize experimentally.

The phase matrices of a four-layer ID$_2$N$_2$ trained using a data set comprising PSF images with pupil functions generated from single

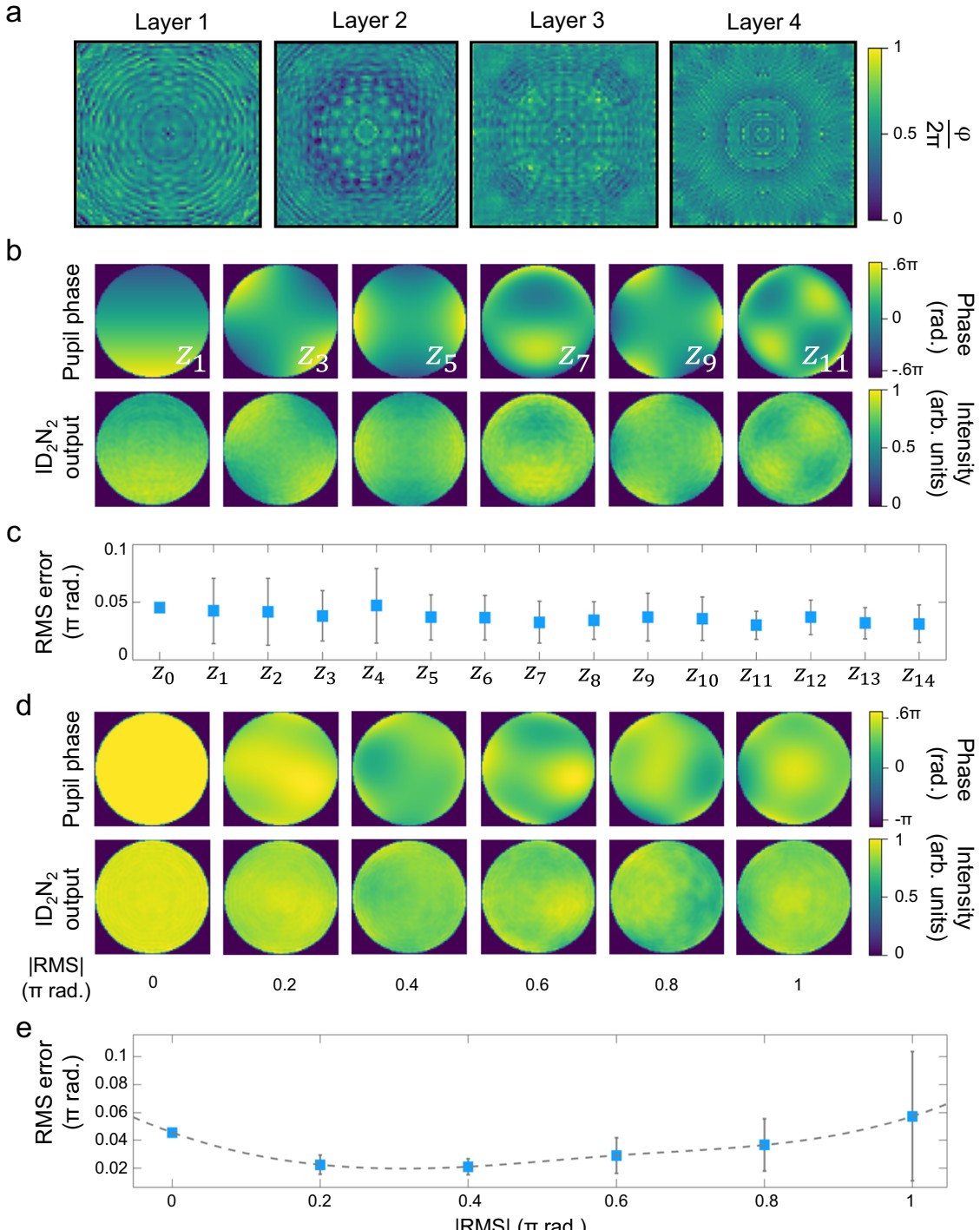

**Fig. 3 | ID₂N₂ response to single and combined Zernike pupil phase testing.** **a** Phase matrices of the trained four-layer DN₂. **b** Pupil phase and simulation results of ID₂N₂ output for six Zernike polynomials. The complete results and performance analysis for the first 14 Zernike polynomials are reported in Supplementary Fig. 3 and Supplementary Fig. 4. **c** Comparison of the root mean square (RMS) error between the original pupil phase and the ID₂N₂ output for the first 14 Zernike polynomials. **d** Imposed and numerically retrieved pupil phases, where the test data set is composed by 1000 randomized pupil phases comprised of linear combination of all the first 14 Zernike polynomials with RMS magnitude from 0 to 1. **e** Comparison of the RMS error between the pupil phases and ID₂N₂ outputs for composed pupil phase with RMS magnitude scanned between 0 and π.

Zernike polynomial from $Z_1$ to $Z_{14}$ (OSA/ANSI indices) is reported in Fig. 3a. Supplementary Fig. 2, the Methods and Supplementary information sections contain the details of the network training.

The ability of this ID₂N₂ to retrieve the pupil phase of a PSFs can be seen in Fig. 3b, Supplementary Fig. 3, Supplementary Fig. 4 and Supplementary Fig. 5, which show the original pupil phases imposed, the resulting PSFs and the corresponding ID₂N₂ outputs with the reconstructed phases. A quantitative evaluation of the ID₂N₂ performance is given by root mean square (RMS) error between the original pupil phase and the ID₂N₂ output. A comparison between performance for the first fourteen Zernike polynomials is reported in Fig. 3c. It was numerically found that the average RMS error is 0.036 π radians.

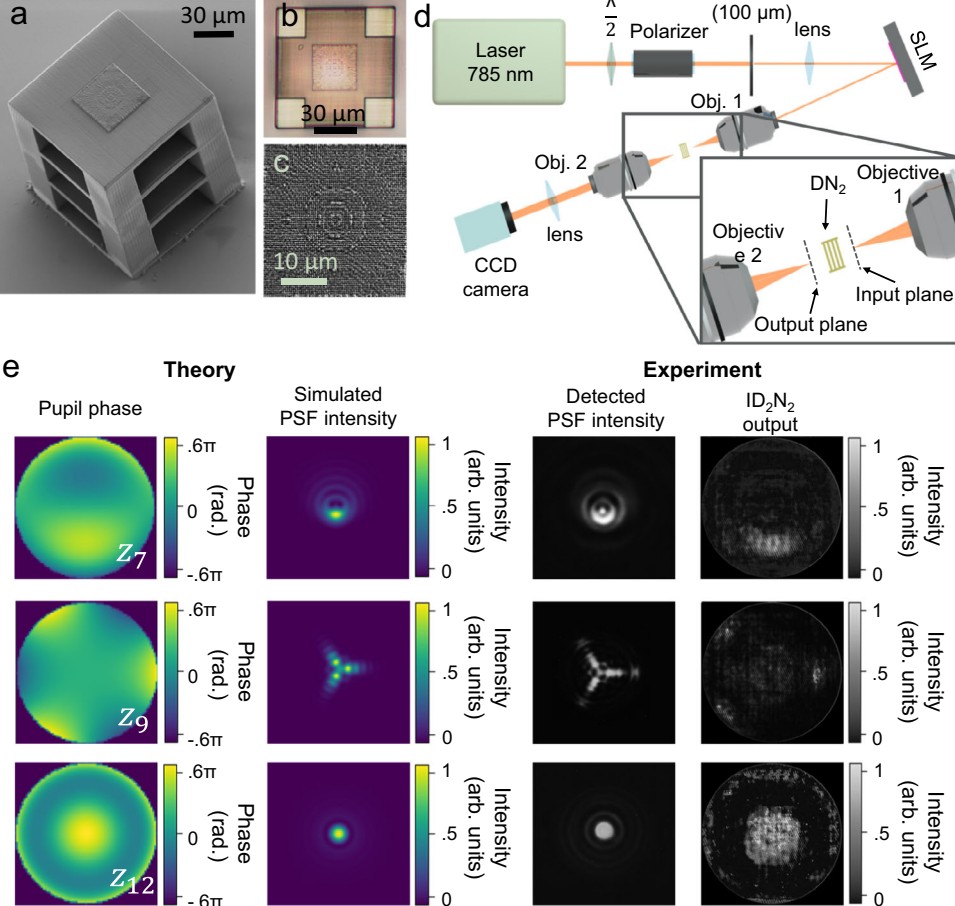

**Fig. 4 | Experimental fabrication and characterization.** Scanning electron microscope (SEM) image (**a**) and optical microscope image (**b**) (top view) of the laser printed $DN_2$. **c** SEM image of a laser printed diffractive element (layer 4). The radius of each pixels is 400 nm, the pixel heights span between 0 and 1.57 μm for a phase modulation 0–2π and the axial nano-stepping is approximately 10 nm[34]. **d** Diagram of the laboratory setup used for testing the $D_2N$. **e** Original pupil phases (with RMS absolute magnitude of 0.6 π radians)

imposed to the PSFs, simulated PSF intensities, experimentally detected PSF intensities and experimentally detected $ID_2N_2$ outputs for the three Zernike polynomials $Z_7$, $Z_9$, $Z_{12}$. The complete results and performance analysis are reported in Supplementary Fig. 9. The images of the experimentally detected PSF and the reconstructed pupil phases consist of 350 × 350 pixels (1.94 × 1.94 mm²). Each $DN_2$ output image is plotted after the application of a 1.3 stretch to better highlight low-intensity features.

Figure 3d, e report the ability of $ID_2N_2$, which was trained with a training data set comprising PSF images generated from single Zernike polynomial pupil phases (Fig. 3a), to retrieve the pupil phase for PSFs generated from Zernike polynomials combinations. Figure 3d shows a comparison between six images of the pupil phase and the corresponding $ID_2N_2$ outputs, whilst Fig. 3e shows a comparison of the RMS error between the imposed and the retrieved pupil phase for PSFs generated from Zernike polynomials combinations with root-mean-squared (RMS) magnitude scanned between ±1π. We also evaluated the performance of an $ID_2N_2$ trained on a data set comprising pupil phases and the corresponding PSFs generated from combinations of Zernike polynomials in reconstructing the pupil phase generated from single and combinations of Zernike polynomials. The results are reported in Supplementary Fig. 6 and Supplementary Fig. 7, showing a worse performance (average RMS error 0.043 π radians) compared to the phase retrieval ability of $ID_2N_2$ trained on a data set comprising PSFs generated from single Zernike polynomial pupil functions.

We evaluated the generalization ability of the proposed $ID_2N_2$ by training the network using two datasets consisting of Zernike polynomials (single Zernike polynomials $Z_1$ to $Z_7$ and single Zernike polynomials $Z_8$ to $Z_{14}$) that were each assigned a random RMS magnitude between ± 0.6 π and their corresponding complex field PSFs. Then, we tested the performance of the network in reconstructing single

Zernike polynomials $Z_1$ to $Z_{14}$. The results reported in Supplementary Fig. 8 show that the networks trained with different subsets of Zernike polynomials have, to some extent, comparable generalization abilities, as confirmed by the average RMS errors calculated for the Zernike polynomials $Z_1$ to $Z_{14}$. However, the prediction ability drops notably when the network predicts polynomials that were not included in the training, as shown by the average RMS errors calculated on the subsets of polynomials.

## $ID_2N_2$ vectorial printing

After calculating the phase delay of each diffractive neuron of the four diffractive layers comprising the $DN_2$ (Fig. 3a), the 3D models of the $DN_2$ were obtained by converting the calculated phase value of each diffractive neuron into a relative height map and then printed in IP-S photoresist using a two-photon nanolithography (TPN) method[34,47]. The results of the TPN printing are reported in Fig. 4a–c.

During fabrication, the position of the polymerizing voxel is controlled with nanometric precision[34,47] in the 3D space thanks to the employment of a TPN system with a piezoelectric nanotranslation stage. Other than commonly used 3D printing methods, where a three-dimensional structure is printed layer-by-layer with discrete height steps in the axial direction, we employ a simplified vectorial printing approach to fabricate the diffractive neurons. Each diffractive neuron

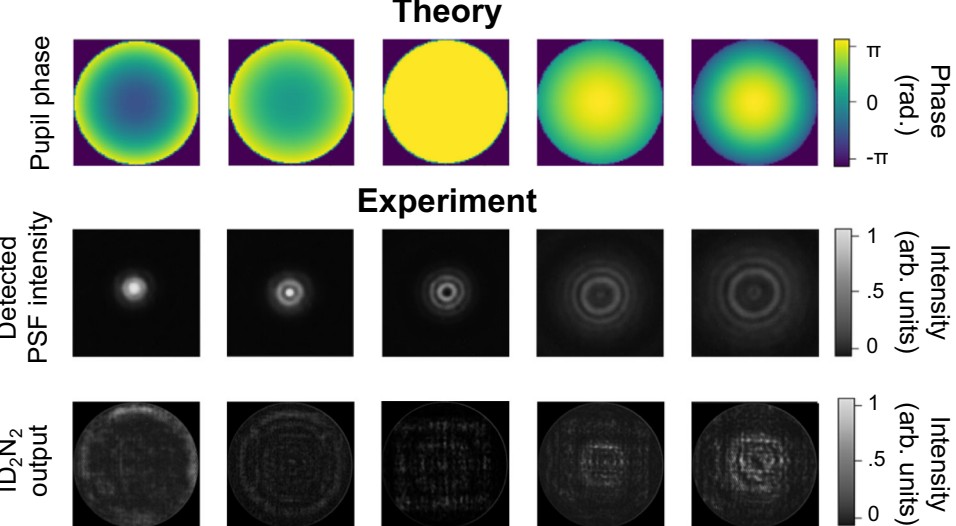

**Fig. 5 | ID$_2$N$_2$ non-degenerate response to defocus.** Defocus (Z$_4$) terms applied and corresponding detected PSF intensities and ID$_2$N$_2$ outputs. The amplitude of the pupil phases is scanned from −π s to +π radians, the PSF are detected at the focal plane of a 10× 0.25NA objective for a point source of 100 μm. Each DN$_2$ output image is plotted after the application of a 1.3 stretch to better highlight low-intensity features.

is printed in a single step as a thin rod that extends into the axial direction, allowing for precise control of the length – and hence phase delay – of each printed diffractive neuron. At the same time, the lateral diameter of the diffractive neurons was precisely controlled through the laser power and the printing speed.

With this method, we were able to achieve diffractive neurons with a lateral diameter of 400 nm, resulting in a neuron density of 625 million neurons per square centimeter, and a nearly continuous phase modulation between 0 and 2π. The Methods and Supplementary information sections contain the details of the TPN setup and the vectorial printing method.

### Experimental performance

To demonstrate the ability of the ID$_2$N$_2$ to retrieve the phase of PSFs and to map it into an intensity distribution, we designed an experimental setup, which, using a coherent laser source and a spatial light modulator (SLM), allowed the projection of PSFs with arbitrary pupil phases at the input plane of the DN$_2$, and the detection of the output with a CCD camera. The experimental layout is shown in Fig. 4d; see the Methods section for a detailed description of the setup and the characterization procedure.

Comparing the pupil phase imposed with the intensity pattern at the output plane of the ID$_2$N$_2$ (Figs. 4e, 5 and Supplementary Fig. 9), we show the ability of these devices to directly reconstruct Zernike-based pupil phase distributions from a PSF. For each measurement in Fig. 4e and Supplementary Fig. 9, a single Zernike polynomial with RMS of 0.6π radians is applied to the SLM. The DN$_2$ output intensities are then recorded with a CCD camera, assuming a uniform response of the detector. The images of the PSFs are also saved for reference.

Qualitatively, the ID$_2$N$_2$ output images prove the ability of the optical network to discriminate PSFs with different pupil phases and to map the pupil phase into an intensity distribution. A quantitative evaluation of the ID$_2$N$_2$ experimental performance is given by the calculation of the RMS error between the original pupil phase and the nanoprinted ID$_2$N$_2$ output reported in Supplementary Fig. 10. For this purpose, we assign values between −0.6 π and 0.6 π to the normalised output intensity pattern reported in Fig. 4 and Supplementary Fig. 9, where the lowest intensity refers to −0.6 π and the highest intensity to +0.6 π and the measured intensity representing the phase in π radians scales linearly with intensity. It was found that the average RMS error is 0.20 with peaks above 0.24 for the third-order Zernike polynomials Z$_{11}$

and Z$_{13}$. The experimental demonstration of the ability of the ID$_2$N$_2$ to map axially symmetric pupil phase is reported in Fig. 5, where the pupil phase of the PSF and the pupil phase predicted by the ID$_2$N$_2$ are shown for defocus terms (Z$_4$) of changing sign and amplitude.

### Four-layer ID$_2$N$_2$ printed on CMOS

In order to harness their complementary physics through integrated on-chip solution, the DN$_2$ were fabricated using the TPN method directly on a CMOS sensor[48] (Fig. 6), realizing in this way a ID$_2$N$_2$ in a single chip. The TPN fabrication method allows the precise fabrication of complex three-dimensional (3D) structures, such as the DN$_2$, on a variety of substrates, with nanometric resolution and without damaging the sensor, essential abilities to access the NIR and VIS wavelength regions and for manufacturing co-integrated opto-electronic systems. Further information on the pretreatments, design and nanoprinting can be found in the Methods and Supplementary information sections. The 3D-nanoprinted CMOS-integrated ID$_2$N$_2$ have the potential to perform in a single step optical inference on complex-valued inputs and digital conversion to reconstruct the pupil phase of the incoming PSFs. The details of the fabrication process and the characterization results for a prototype device with limited functionality are shown in Supplementary Fig. 11, Supplementary Fig. 12 and in the CMOS prototype section of the Supplementary Information.

## Discussion

The ID$_2$N$_2$ presented in this work combines four diffractive optical elements with the nonlinear detection through an imaging sensor, achieving a hybrid opto-electronic integrated deep neural network with phase retrieval capabilities. This compact device operates in the NIR wavelength region and processes complex field PSFs through optical inference, performing direct retrieval of the incoming field pupil phase, including the magnitude and the sign. Numerical simulations validate the principle of the ID$_2$N$_2$, and laboratory demonstrations confirm its performance. Our goal throughout this work was not to match the highest accuracy achieved with a state-of-the-art ultra-deep CNN trained for phase retrieval tasks[49,50], but rather to understand and characterize the behavior of the ID$_2$N$_2$ when interacting with complex fields, proofing the ability to retrieve phase information and assess its potential in a hybrid optoelectronic architecture.

In the optoelectronic network architecture presented here, the ReLU is applied as last output layer that adds a bias term to the output

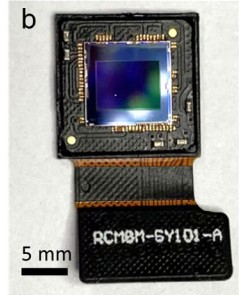
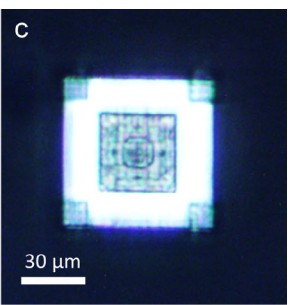

**Fig. 6 | Multilayer diffractive neural networks integrated on CMOS.** The nanoscale DN2 is physically 3D printed by TPN method (**a**) directly on the CMOS sensor. **b** Photograph of the Sony IMX219 NoIR CMOS sensor with a $DN_2$ 3D printed via the TPN method. **c** Optical microscope image (top view) of the laser printed $D_2N$ (top view) printed on the CMOS sensor.

image – i.e. changes the value of all pixels by a constant, adjusting the brightness of the image – improving the performance of the network (Supplementary Fig. 1b and Supplementary Fig. 1c). The application of the nonlinear activation function as last layer of the network constitutes an architectural difference compared with computer-based neural networks.

However, the $ID_2N_2s$ could be incorporated in a more complex optoelectronic scheme, where the response of the detector module would play the role of the first nonlinear hidden layer of the network[33,51,52].

The $ID_2N_2$ presented in this work are trained to operate with temporally and spatially coherent light at a 785 nm wavelength. The use of other wavelengths or spatially and temporally incoherent illumination[37,53] on the networks presented in this work, will result in unpredictable outputs.

In numerical experiments, we introduced pupil phases with RMS magnitude between ±0.6π radians constructed using the first 14 Zernike polynomials individually and in combination, and we verified that the pupil phases of the PSFs in input are accurately reconstructed from the $ID_2N_2$ with an average precision of 0.036 and 0.043 π radians RMS error for the single and combined Zernike polynomials, respectively. In either case, the lowest RMS error is achieved for RMS absolute magnitude between 0.2 and 0.4 π radians. The tendency of PSFs to approach a uniform diffraction-limited shape means that the prediction ability of the $ID_2N_2$ is limited for RMS absolute magnitudes lower that 0.2 π radians. For RMS absolute magnitudes above 0.6 π radians, the challenge is to reconstruct pupil phase images with higher contrasts. These challenges may be mitigated by using a lager training dataset[11] or by training the $ID_2N_2$ only on the specific range of RMS magnitudes for which it is designed to encounter in the optical system.

In our optical experiments, we characterized the performance of the 3D nanoprinted $ID_2N_2$ finding that single polynomials pupil phases of the PSFs in input are reconstructed with an average precision of 0.191 π radians RMS error. Several factors might have contributed to the discrepancies observed between our numerical simulations and the experimental results reported. Sources of errors could be any lateral and axial misalignments between the diffractive layers due to shrinking and distortions that might be caused by capillary forces arising during the evaporation of the developer and rinsing liquids. Moreover, the incident light field is assumed to be spatially uniform and to propagate parallel to the optical axis of the $DN_2$. Additional experimental errors might be introduced in our results due to the imperfect beam profile and alignment with respect to $DN_2$. Finally, potential fabrication inaccuracies (errors in the printing of single diffractive neurons) or impurities in the photoresist could also contribute with additional errors in our experimental results compared to the numerical results.

The experimental characterization of the $ID_2N_2$ was performed using a CCD camera, while the integration of $DN_2$ with a CMOS

detector module aimed to prove the integrability of our optoelectronic framework. In our current method for designing the $ID_2N_2$, all the layers of the network (i.e. input layer, diffractive layers, output layer) have the same number of pixels and the same pixel pitch. In our case the pixel size and pitch were chosen to maximize the performance of the $DN_2$ for the chosen operative wavelength (785 nm) without considering the pixel size/pitch of the CMOS sensor. For an optimised design that consider the features of the CMOS sensor, several solutions can be considered. For example, the design of the diffractive layer could be adjusted to match the CMOS pixel size. This would result in diffractive neurons with large diameter compared with the operative wavelength, and consequently a low diffraction efficiency and an increased distance required between layers to form fully connected layers. Alternatively, the method for training the $DN_2$ could be extended to consider layers with different numbers of pixels, for example, through changing the sampling of the fields after propagation between the respective layers.

Other than a spatial mode sorter that implements a modal-based approach to phase retrieval, our method is a physical implementation of a direct phase retrieval scheme, where the $ID_2N_2s$ map a light beam into an intensity distribution representing the beam's Zernike-based pupil phase, an image that gives information on the beam's original aberration. However, the $ID_2N_2$ can be trained to perform spatial mode sorting, multiplexing and de-multiplexing of light beams, following a modal-based approach[42,54].

In conclusion, the $ID_2N_2$ module presented in this work may be used as a compact phase sensing element co-integrated on off-the-shelf CMOS imaging technology that can retrieve a pupil phase in a single step. We have shown the capability of the $ID_2N_2$ to directly retrieve the pupil phase of superpositions of the first fourteen Zernike polynomials through passive optical inference in conjunction with a nonlinear optical-electronic conversion step. The $ID_2N_2$ presented here allows for direct retrieval of Zernike-based pupil phase, although the error rates may be too high for certain applications. To overcome this limitation, the $ID_2N_2$ can be used as a building block in larger scale optoelectronic deep neural network, where the nonlinear response of the detector module can act as a hidden nonlinear layer. Also, the output of the $ID_2N_2$ can be considered as an optically pre-processed entry point to digital phase retrieval algorithms. The combination of $ID_2N_2$ with electronic networks will increase the performance of phase retrieval systems by leveraging the advantages of optical pre-processing with a diffractive network, such as the ability to discriminate the sign of a pupil phase, and reduce the requirements for computational resources and multiple sensors.

Hybrid optical-electronic solutions, such as the one presented in this work, may form the basis of new highly integrated $ON_2$-based systems to perform inference tasks on information native to the optical domain faster, more efficiently and more robustly than conventional digital neural networks. Consequently, this development

might have transformative impact on aberration correction with adaptive optics, data processing[55] and sensing[56] and may be crucial in the development of robust and generalized quantitative phase imaging methods[1] with low computational complexity and memory footprint, to be applied, for example, in biological cell- and tissue imaging problems.

## Methods

### In silico training

The $ID_2N_2$ is modelled on a computer combining a multi-layer $DN_2$ (input, four diffractive layers and output) with a ReLU nonlinear activation function (Fig. 2a). Each layer of the $DN_2$ consisting of $N \times N$ resolvable pixels that act as diffractive neurons, which receive, modulate and transmit a light field. The diffractive neurons of each layer are linked to the diffractive neurons of the neighboring layers through Rayleigh-Sommerfeld[57] diffraction. While the diffractive neurons of the input and output layers are unbiased (i.e., uniform), each neuron of a diffractive layer adds a bias in the form of a phase delay to the transmitted signal. The phase delay, modelled as the complex transmission coefficient of each pixel can be considered as a learnable parameter that can be iteratively adjusted during the computer-based training. A mean-squared-error loss function[58] is defined to evaluate the performance of the $ID_2N_2$ with respect to the targeted application, and an algorithm iteratively optimises the phase delay of each diffractive neuron in the diffractive layer to minimise the loss function. We achieve the MLD design using the TensorFlow (Google Inc.)[59] framework, used to implement a forward propagation model. We employ the stochastic gradient descent algorithm Adam[60] to back-propagate[61] the errors and update the diffractive layer phase parameters to minimise the loss function. The desired mapping functions between the input and output planes are achieved after 500 epochs and an empirically optimized learning rate of 0.0001. The model is implemented using Python version 3.5.0 and TensorFlow framework version 1.4.0 (Google Inc.). The Supplementary information section contains further details of this TensorFlow-based design and training processes.

### Training and test dataset processing

To build the training dataset random pupil phase functions were generated using Zernike polynomials from $Z_1$ to $Z_{14}$ (OSA/ANSI indices) as they covered the strongest aberrations typically seen in microscopy. The corresponding PSFs were generated by a fast Fourier transform (FFT) implementation of vectorial Debye theory[44]. The optical system chosen for our training consisted of infinity corrected objective lens with a numerical aperture (NA) of 0.25, a magnification of 10× operating in air and a point source of 100 μm. A coherent, monochromatic wave field with uniform amplitude distribution parallel to the optical axis of the simulated objective crossing the aperture stop (entrance pupil) was considered in the current experiments. The PSF was calculated at an axial offset from the geometric focus by a distance of +20 μm. For the optical system detailed above, a complete image of the PSF requires a lateral dimension of approximately 30 μm, that can be divided in 75×75 diffractive neurons with a diameter of 400 nm. The wavelength of the point source was 785 nm. The size of the diffractive layers has been chosen to match the size of the PSF generated by the 10× NA0.25 objective and the diffractive neuron size and number have been optimised for the chosen wavelength. The separation between the $DN_2$ layers is 40 wavelengths (31.4 μm). The numerical calculations of the PSF were validated by the comparison between the numerically calculated PSF intensities and the experimentally detected PSF intensities reported in Fig. 4e and Supplementary Fig. 9.

In our numerical experiments we train an $ID_2N_2$ using a dataset consisting of 8000 randomly selected single Zernike polynomials between $Z_1$ and $Z_{14}$ that were each assigned a random root-mean-square (RMS) magnitude between ±0.6 π and the corresponding complex field PSF. We than test the ability of the same network to reconstruct:

-   single Zernike polynomial pupil phases (Fig. 3c) using a test data set comprising 800 PSFs with pupil phases generated by each of the fourteen Zernike polynomials used in the training data set. For this test, the random root-mean-square (RMS) magnitude of each individual polynomial was varied between ±0.6π radians such that the entire training range was probed.
-   combinations of Zernike polynomial pupil phases (Fig. 3e) using a test data set comprising 8000 PSFs with pupil phases generated using linear combination of Zernike polynomials from $Z_1$ to $Z_{14}$. The magnitude of each individual pupil phases was varied between ±1π radians and the combinations were then normalized to ±0.2π, ±0.4π, ±0.6π, ±0.8π and ±1π radians to study the response to different aberration magnitudes.

The non-degenerate response of the network to defocus, was evaluated using training and test data sets generated by each of the fourteen Zernike polynomials, where the random root-mean-square (RMS) magnitude of each individual polynomial was varied between ±1π radians.

### Nanoprinting

Polymeric $DN_2$ are printed by a TPN[47,62] method based on femtosecond laser pulses and two-photon absorption. A femtosecond fiber laser (Spectra Physics, InSight X3) provides laser light at a wavelength of 800 nm. The laser pulses with a width of 120 fs and a repetition rate of 80 MHz are steered by a 4f imaging system into a 1.45 NA × 100 oil immersion objective (Olympus). A piezoelectric nanotranslation stage (PI P-545.xC8S PInano Cap XY(Z) Piezo System, Physik Instrumente) is used to trace out the microstructures in the photoresist. We employ a dip-in TPN approach using commercial IP-S (Nanoscribe GmbH) photoresist. To ensure the proper distance between the different layers of the $DN_2$, we designed a frame structure comprising layer supports on pillars (Fig. 4a).

We manufacture $ID_2N_2$ on a Sony IMX219 NoIR CMOS image sensor from a Raspberry Pi Camera Module (Fig. 6). Before manufacturing, we remove the lenses and clean the sensor surface with ethanol. After the TPN procedure, the sample is developed in propylene glycol methyl ether acetate (PGMEA, 1-methoxy-2-propanol acetate/ SU-8 developer) for 20 min, rinsed with isopropanol, ethanol, and then dried at room temperature.

### Characterization setup

A schematic diagram of the experimental setup is given in Fig. 4b. The light beam is generated through a Thorlabs OBIS 785 nm coherent laser source. The point source is generated by a 100 μm pinhole. The polarised beam is then directed on a Hamamatsu SLM X13138-07 (620–1100 nm) and through a 10× Olympus objective (0.25 NA), is focused at the input plane of the $ID_2N_2$ at 20 μm from the objective focal point. After passing through the $ID_2N_2$, the output image is collected by a CCD camera (Basler ace acA2040-90uc, frame rate 90 Hz).

## Data availability

All data needed to evaluate the conclusions in the paper are present in the paper and Supplementary Information. Additional data related to this paper may be requested from the corresponding author, E.G.

## Code availability

The custom code and mathematical algorithm used to obtain the results within this paper may be requested from the corresponding author, E.G.

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

## Acknowledgements

The authors acknowledge the funding support from the Shanghai Natural Science Foundation (21ZR1443400), Shanghai Rising-Star Program (21QA1403600). Min Gu would like to acknowledge the support from the Science and Technology Commission of Shanghai Municipality (Grant No. 21DZ1100500), and the Shanghai Frontiers Science Center Program (2021-2025 No. 20).

## Author contributions

E.G. conceived the concept, and M.G. proposed this idea and supervised the project. E.G. and S.S. performed numerical simulations, T.P.N. experiments and experimental characterisation. All authors participated in discussions and contributed to writing of the manuscript.

## Competing interests

The authors declare no competing interests.
