## [Peer Review File · Nature Communications]

REVIEWER COMMENTS

Reviewer #1 (Remarks to the Author):

The paper describes a novel method for sensing of aberrations in wavefronts using a combination of multi-element diffractive gratings and machine learning. The diffractive elements are made in three dimensions using two-photon laser nanofabrication, designed through machine learning/diffraction methods. This is an interesting and innovative combination, that brings something new to the field. The method is effective, although still at an early stage. I can imagine this concept could be extended somewhat, so it is likely to spur follow on ideas, which would be useful for the field. The optical and processing concepts are shown to be effective within the range of operation specified, which is small, but sufficient to show the concept. There is a demonstration of how the diffractive element could be produced directly on a CMOS imaging chip, although this demonstration does not seem to be operational. I can imagine that there are considerable technical issues in implementing a fully functioning combination of these technologies in a lab, which would be solved without problem in an industrial fabrication setting. Overall, with some changes as outlined below, I think this can be published.

Points to be addressed by the authors:

- It is mentioned that the detector non-linearity is modelled as a ReLU function. This non-linearity is an important part of the machine learning functionality. It is explained that this arises from the “non-linear response of the CMOS detector”. Has the response of the detector been measured? What is the physical basis for choosing the ReLU function?
- As there is non-uniformity of response across CMOS arrays, there is often a lot of post processing required. Has this been taken into account?
- In some places the detector is described as “CMOS” and in other places as “CCD”. Which is it? They are not the same.
- Some text is struck out in the caption of Fig 5.
- In section 2.5, it is implied that the diffractive element on CMOS device is not functional, through the phrase “[they] have the potential to perform in a single step optical inference ...”. However, the abstract includes the sentence “The integration of the diffractive elements on CMOS sensors allows for the direct extraction of the pupil phase information from the detector without additional digital post-processing”, which implies that the system has been implemented. The abstract should be more carefully worded.

Reviewer #2 (Remarks to the Author):

The authors present a diffractive optical neural network in a compact form factor and proposed to use it for direct pupil aberration estimation. While the demonstration of a compact DNN on a CMOS sensor is novel and impressive, I do have some major concerns about the claims and their fundamental limitations for using it for phase retrieval. Moreover, to me, this work seems to be a natural extension and combination of their two previous works from the authors – a fabrication method reported in Ref. [36] and an algorithmic framework from Ref. [12]. I do not find other untrivial ideas or promising potential that this system could scale up or be improved for direct phase retrieval of arbitrary phases. The manuscript lacks the experimental results from the compact prototype, and all the results are demonstrated with the benchtop prototype (please correct me if I'm wrong). Thus, I cannot recommend publication in Nature Communications, at least in its current form.

First of all, I think “direct phase retrieval” is a bit overclaimed because all the results presented here are the estimation of Zernike-based wavefronts, which is relatively smooth and usually used for aberration estimation (as they also mentioned themselves “using Zernike polynomials from Z1 to Z14 (OSA/ANSI indices) as they covered the strongest aberrations typically seen in microscopy”). I don't see any clue that this system can be used for retrieving more complex phase images other than these aberrations from the manuscript. Thus, in my opinion, they should tone down their claim to “direct estimation of aberration functions” like in Ref. [12], unless they can demonstrate a promising phase retrieval result at arbitrary planes (not pupil plane) or arbitrary examples such as from biology samples. For example, they claim “the pupil phase of incoming PSFs with low error rates” which is relatively not consistent in my opinion with their numbers of 0.036pi (in simulation) or 0.191pi (in experimental) on average. This is too high to retrieve arbitrary pupil phase, probably only be able to observe the tendency of the magnitude or sign as they discuss.

Second, the authors claim that “This compact device operates in the NIR wavelength region and processes complex field PSFs through all-optical inference, performing direct retrieval of the incoming field pupil phase, including the magnitude and the sign.”. But I don't think their system is “all-optical” since they explicitly mention “optical-electronic conversion step” for nonlinear activation, which is not clear how to be characterized nor how efficient, while they mention 90Hz as their sensor refresh rate. It is very unclear how this nonlinear activation function is implemented. While the authors claim “The nonlinear activation function is implemented during the photoelectric conversion via the imaging sensor.” and model it with some ReLU function, it would be great to

clarify which nonlinearity they are referring to: (is it just the photoelectric effect or some Gamma/square curves?)

My other concern is that the DNNs might not have the capability to retrieve arbitrary phases, due to their inherent linearity. Then “nonlinearity at last” scheme mentioned in the previous paragraph is actually more problematic since it makes it unclear how to increase the system’s capability although the authors set the scope of the work as “Our goal throughout this work was not to match the highest accuracy achieved with a state-of-the-art ultra-deep CNN trained for phase retrieval tasks^{49,50}, but rather to understand and characterize the behavior of the ID2N2 when interacting with complex fields, proving the ability to retrieve phase information and assess its potential in a hybrid optoelectronic architecture.”. I think the performance wouldn’t scale with the number of diffractive layers since they are all linear, which makes it challenging to retrieve the arbitrary (pupil) phase as they claim. It would be great if the authors could add analysis on the capacity of the system, or the scalability of its non-linearity.

Additionally, the following might be confusing to readers:

- In the abstract, they claim an average RMS error of 0.036 pi radians but I guess this is in numerical simulation, not the experimental results. This should be clarified.
- It is confusing if this system can fully support 2pi range retrieval or those are scaled from 1.2pi range, since Fig. 2 shows 2pi range colorbars while the manuscript does mention 1.2pi range. I guess it is stretched somehow and the network itself has limited capability of supporting 2pi range though, this should be clarified.
- I do see strikethroughs on the optimised ReLU function, what are they?
- The TPN should be expanded when it is introduced first.
- In lines 391-399, “The performance of the network in reconstructing combinations of Zernike polynomial pupil phases was evaluated using a test data set comprising 8000 PSFs with pupil phases generated using a linear combination of Zernike polynomials from Z1 to Z14. The magnitude of each individual pupil phase was varied between $\pm 1\pi$ radians to study the response to different aberration magnitudes” This paragraph is just repeated with slightly different numbers.

In this paper, Goi et al demonstrated a 4-layer diffractive neural network fabricated by two-photon polymerization that can convert intensity point spread function to intensity distribution that is proportional to the pupil phase that resulted in the point spread function. I agree with the authors that real-time analysis of optical wavefront is a promising application of diffractive neural networks, and the paper took a systematic approach with well executed experiments to prove the practicality of this method for NIR wavelengths. However, there are several key technical aspects that need to be clarified in order to enable readers to truly evaluate the potential of the method:

1. What is the role of ReLU layer in the CCD or CMOS camera? Since nonlinear layers only serve to improve the performance of neural networks when they function as hidden layers instead of the last output layer and the authors claim that no more digital operation was needed after reading out the light intensity distribution from the camera, it is not very clear how a shifted (or so-called optimised) ReLU function improves the performance other than adding a bias term to the image. The claim of ReLU nonlinear function can be potentially misleading for the readers to believe the nonlinear operation contributes to the computation as those in deep neural networks, which is not true in this study.

2. Is the mapping from the pupil phase to intensity PSF a linear one? if so, the author should show the derivation. If not, how could the spatial modes be transformed by a diffractive neural network, which performs linear operations? if the authors believe the diffractive neural network performs nonlinear operation to some capacity, say with quadratic nonlinearity from amplitude to intensity detection, the authors should also carefully discuss this point. I say “to some capacity” since the quadratic nonlinearity, even as hidden layers, does not make the diffractive neural network a universal approximator, so it does not give the full benefit of nonlinearity to the neural networks. The authors are welcome to make arguments on this point.

3. Would the authors elaborate more on the following statement?

“It is important to note that an ID2N2 trained using only the intensity pattern of the PSFs is not able to distinguish the sign of defocus, while the ID2N2 trained using the complex fields that describe the PSFs can retrieve the information contained within the phase structure of the PSF and encode it in the intensity of the outputs²⁴”

Does it mean the intensity measured at plane 1 in Figure 4d and e cannot be used directly for training the diffractive neural network? Instead either the phase of the PSF also needs to be measured at plane 1 or a phase pattern needs to be assumed for each measured PSF in order for the training to work well?

4. What is the real generalization ability of the diffractive neural network for Zernike polynomials? Even though the study used differently randomly generated linear combination of Zernike polynomials for train and test respectively, they were still trained on the same basis of Zernike polynomials. Essentially, I wish to know, for example, if the train set contains Z_1 to Z_7 ,

and the test set contain Z8-Z14, would the training result generalize to test? In a more fundamental way, the question is about whether the linear operation of the diffractive neural network is truly able to model a physical system of such complexity, and whether it will become much more accurate with further developments.

5. My understanding is that the authors assumed a uniform amplitude distribution at the pupil plane, and then only seek to recover phase pattern at the pupil plane. What if there is actually an intensity distribution at the pupil plane, is it possible to infer the amplitude AND phase of the field based on the intensity PSF? Have the authors performed simulation on this? Please comment on it.

6. The authors should discuss the difference and similarities of their method from spatial mode sorting with multiple diffractive elements, for example, demonstrated in the following paper with multi-plane light conversion: <https://doi.org/10.1038/s41467-019-09840-4> It is not very clear to me how the demonstrated experiment is more than a mode sorter that maps a linear combination of a set of spatial modes to a linear combination of another set of modes with the same linear coefficients.

Other technical issues:

1. Why does the phase plate appear to be axially symmetric in Figure 2a? Were they trained exclusively with central symmetric Zernike patterns?

2. I am curious why the diffractive neural network output trained with complex field in Figure 2c has more fine structures compared to Figure 2b trained with intensity field. Are those feature or artifacts?

3. There seems to be inconsistency on which camera is used for detection and implements nonlinearity. CMOS camera is mentioned in section 2.5 and Figure 4, but CCD camera is mentioned everywhere else for nonlinear ReLU functions. Is the nonlinear function really caused by “photoelectric conversion” as claimed by the authors, or just some analog circuit setting? Regardless of these details, I respectfully disagree with the claim that ReLU nonlinear functions play any significant role in this experiment.

4. The following statements are confusing:

“Each neuron is printed in a single step as a thin rod that extends into the axial direction, allowing for precise control of the length – and hence bias – of each printed artificial neuron.”

“While the neurons of the input and output layers are unbiased (i.e., uniform), each neuron of a diffractive layer adds a bias in the form of a phase delay to the transmitted signal.”

This is because it is a convention in machine learning community that the bias of artificial neurons refer to the b term in the update equation of $Wx+b$. However, what the ‘bias’ actually means here is the *phase* bias, which has a completely different meaning. Since there is really no

one-to-one analogy between the mathematical expression of diffractive neural network and digital neural networks (see Figure 1D of <https://doi.org/10.1126/science.aat8084>), using the same term (e.g., artificial neurons, bias) to address different mathematical entities in the two models only serve to confuse readers. The authors should try to make clear and unambiguous statements.

5. I strongly encourage the authors to share their simulation code by depositing in a publicly available online repository.

6. My understanding is that a coherent laser source was used for this experiment. The authors should clearly state the demonstration was made with coherent light, and it is preferred to have a discussion on what would happen if the light source is incoherent.

Minor points:

1. “Matrixes” should be “matrices”.
2. Citation 1 and 2 are the same
3. The author list of citation 30 seems to be wrong

Please find below the requested responses to the Reviewers comments for our manuscript “Direct retrieval of Zernike-based pupil functions using integrated diffractive deep neural networks”, originally submitted with the title “Direct phase retrieval using integrated diffractive deep neural networks” (NCOMMS-22-23143). We color-coded this reply as follows:

- the original comments of the Reviewers are written in black
- our responses to the Reviewers comments are written in blue
- the changes to the manuscript are written in green

All the changes in the manuscript text file are highlighted in yellow.

REVIEWER COMMENTS

Reviewer #1 (Remarks to the Author):

The paper describes a novel method for sensing of aberrations in wavefronts using a combination of multi-element diffractive gratings and machine learning. The diffractive elements are made in three dimensions using two-photon laser nanofabrication, designed through machine learning/diffraction methods. This is an interesting and innovative combination, that brings something new to the field. The method is effective, although still at an early stage. I can imagine this concept could be extended somewhat, so it is likely to spur follow on ideas, which would be useful for the field. The optical and processing concepts are shown to be effective within the range of operation specified, which is small, but sufficient to show the concept. There is a demonstration of how the diffractive element could be produced directly on a CMOS imaging chip, although this demonstration does not seem to be operational. I can imagine that there are considerable technical issues in implementing a fully functioning combination of these technologies in a lab, which would be solved without problem in an industrial fabrication setting. Overall, with some changes as outlined below, I think this can be published.

Response:

We are glad about the Reviewer’s positive comments on our work, as well as the recommendation for publication. We also thank the Reviewer for their constructive feedback. We addressed their comments in the revised manuscript as described in detail below.

Points to be addressed by the authors:

- It is mentioned that the detector non-linearity is modelled as a ReLU function. This non-linearity is an important part of the machine learning functionality. It is explained that this arises from the “non-linear response of the CMOS detector”. Has the response of the detector been measured? What is the physical basis for choosing the ReLU function?

Response:

We thank the Reviewer for the question that allows us to clarify our presentation.

For this experimental demonstration of the ID₂N₂ performance, we did not perform a characterization of the detector module response. The choice to use the ReLU function to approximate the nonlinearity of the detector is motivated by the response curve for a light sensitive detector module consisting of

photoelectric conversion and electronic readout circuit, which can be divided into three parts: the dark area, the linear area and the saturation area, as shown in the figure below. The dark area of the response curve shows the detector modules response to very low light intensities; the output value of the detector module in this area is noisy and unpredictable. After the Noise Equivalent Exposure (NEE) point is reached, the output of the detector module becomes linear until the Saturation Equivalent Exposure (SEE) is reached. At this point, increasing the light intensity results in a nonlinear increase in the detector module output. Our experiments happen before the SEE point and therefore the response function can be approximated by a ReLU function.

Figure: A typical response curve of a light sensitive detector module. Image adapted from <https://www.baslerweb.com/en/sales-support/knowledge-base/frequently-asked-questions/what-is-sensitivity-and-why-are-sensitivity-statements-often-misleading/14987/>.

To clarify this point we added:

- The following sentence to the first paragraph of the *Results* section of the manuscript: ‘The choice to use the ReLU function to approximate the nonlinearity of the detector is motivated by the response curve for a light sensitive detector module consisting of photoelectric conversion and electronic readout circuit (see Supplementary Information)’.
- The above discussion on the ReLU function to the *TensorFlow-based design and training* section of the Supporting Information.

As there is non-uniformity of response across CMOS arrays, there is often a lot of post processing required. Has this been taken into account?

Response:

As the Reviewer highlighted in her/his initial statement, the detector used in the experimental characterization of the ID₂N₂ is a CCD camera. In the submitted manuscript we did not demonstrate the functionality of the diffractive network printed on CMOS, therefore we did not characterize experimentally the non-uniform response of the CMOS sensor.

The size of our diffractive networks is 30×30 μm, which corresponds to 27×27 pixels on a Sony IMX219PQ CMOS image sensor with pixel size 1.12×1.12 μm². Given that the nonuniformity of a CMOS array typically changes slowly across the array, we would not consider this as a significant source of error in the current configuration. However, if the lateral size of the diffractive elements were to be scaled up in an industrial fabrication context, this nonuniformity would certainly need to be considered in post processing.

To clarify this point, we added the following sentences:

- in the *Experimental performance* section of the manuscript:
‘The DN₂ output intensities are then recorded with a CCD camera, assuming a uniform response of the detector’.
- in the *Discussion* section of the manuscript:
‘The experimental characterization of the ID₂N₂ was performed using a CCD camera, while the integration of DN₂ with a CMOS detector module aimed to prove the integrability of our optoelectronic framework. The size of the diffractive elements presented here (30×30 μm²) corresponds to 27×27 pixels on a Sony IMX219PQ CMOS image sensor with pixel size 1.12×1.12 μm², these images would be too low in resolution to appreciate the functionality of the optical network. The minimum number of pixels on the CMOS sensor required to evaluate the functionality of the network would be 75×75 pixels, which is the resolution of the training datasets. This would require diffractive elements sized 84×84 μm², which is not only challenging to achieve with the TPN method in a laboratory environment due to long fabrication times, but would also require to consider nonuniformities in CMOS detector response during a post-processing step.’

- In some places the detector is described as “CMOS” and in other places as “CCD”. Which is it? They are not the same.

Response:

The detector used in the experimental characterization of the ID₂N₂ is a CCD camera. Therefore, in the *Experimental performance* and in the *Methods, Characterization setup* sections of the manuscript, we referred to the CCD camera. We printed an ID₂N₂ on a CMOS detector to demonstrate the integrability of these optical networks with commercial CMOS chips.

We added a clarification in the last paragraph of the *Discussion* section of the manuscript, this has also been included in the response to the Reviewers previous question for context:

‘The experimental characterization of the ID₂N₂ was performed using a CCD camera, while the integration of DN₂ with a CMOS detector module aimed to prove the integrability of our optoelectronic framework.

- Some text is struck out in the caption of Fig 5.

Response:

We thank the Reviewer for pointing out this oversight. We corrected it in the main text.

- In section 2.5, it is implied that the diffractive element on CMOS device is not functional, through the phrase “[they] have the potential to perform in a single step optical inference ...”. However, the abstract includes the sentence “The integration of the diffractive elements on CMOS sensors allows for the direct extraction of the pupil phase information from the detector without additional digital post-processing”, which implies that the system has been implemented. The abstract should be more carefully worded.

Response:

The Reviewer states correctly that in the submitted manuscript we did not demonstrate the functionality of the diffractive network printed on CMOS for reasons addressed in the reply to the Reviewers previous comments, and that the abstract is therefore misleading. We rephrased the last sentence of the *Abstract* as follows:

‘The integrability of the diffractive elements with CMOS sensors shows the potential for the direct extraction of the pupil phase information from a detector module without additional digital post-processing.’

Reviewer #2 (Remarks to the Author):

The authors present a diffractive optical neural network in a compact form factor and proposed to use it for direct pupil aberration estimation. While the demonstration of a compact DNN on a CMOS sensor is novel and impressive, I do have some major concerns about the claims and their fundamental limitations for using it for phase retrieval. Moreover, to me, this work seems to be a natural extension and combination of their two previous works from the authors – a fabrication method reported in Ref. [36] and an algorithmic framework from Ref. [12]. I do not find other untrivial ideas or promising potential that this system could scale up or be improved for direct phase retrieval of arbitrary phases. The manuscript lacks the experimental results from the compact prototype, and all the results are demonstrated with the benchtop prototype (please correct me if I'm wrong). Thus, I cannot recommend publication in Nature Communications, at least in its current form.

Response:

We thank the Reviewer for the time dedicated to review our work and the comments she/he provided. We hope to be able to address the Reviewer's concerns regarding our manuscript in our response.

We are glad that the reviewer appreciated the novelty of our demonstration and would like to take this opportunity to better highlight the novelty of our work. The numerical and experimental demonstration that a single compact device consisting of a diffractive neural network and a camera sensor can indeed directly reconstruct Zernike-based pupil phases over 2D maps through optical inference, is conceptually new and here lies the impact of our work. The integration of DN_2 with a CMOS detector module aimed to prove the integrability of our optoelectronic framework, rather than achieving a functional prototype. A detailed discussion on this can be found in the reply to Reviewer #1, question 2.

First of all, I think “direct phase retrieval” is a bit overclaimed because all the results presented here are the estimation of Zernike-based wavefronts, which is relatively smooth and usually used for aberration estimation (as they also mentioned themselves “using Zernike polynomials from Z1 to Z14 (OSA/ANSI indices) as they covered the strongest aberrations typically seen in microscopy”). I don't see any clue that this system can be used for retrieving more complex phase images other than these aberrations from the manuscript. Thus, in my opinion, they should tone down their claim to “direct estimation of aberration functions” like in Ref. [12], unless they can demonstrate a promising phase retrieval result at arbitrary planes (not pupil plane) or arbitrary examples such as from biology samples. For example, they claim “the pupil phase of incoming PSFs with low error rates” which is relatively not consistent in my opinion with their numbers of 0.036π (in simulation) or 0.191π (in experimental) on average. This is too high to retrieve arbitrary pupil phase, probably only be able to observe the tendency of the magnitude or sign as they discuss.

Response:

Motivated by the Reviewer #2's concern raised in this comment and by Reviewer #3's question about the generalization capability of our ID_2N_2 framework, we run numerical experiments to test the network ability to generalize the predictions and to retrieve arbitrary phases after being trained with single Zernike polynomials. The results show that, while there is a certain ability to generalize (as discussed in detail in the response to Reviewer #3, question 4), the reconstruction of arbitrary pupil phases cannot be achieved by the ID_2N_2 in the current training configuration.

Therefore, we agree with the Reviewer that the claim of direct phase retrieval of arbitrary complex phase images needs to be rephrased and the applicability of the framework need to be discussed more in detail.

To address the points raised by the Reviewer, we performed the following changes in the manuscript:

- “Direct phase retrieval” was changed in “direct retrieval of Zernike-based pupil phase” in the title and in the manuscript.
- Removed “low error rates” from *Abstract* and *Conclusions* section of the manuscript.
- Added the following discussion on the applicability of the method in the *Conclusion* section of the manuscript:

‘The ID₂N₂ presented here allows for direct retrieval of Zernike-based pupil phase, although the error rates may be too high for certain applications. To overcome this limitation, the ID₂N₂ can be used as a building block in larger scale optoelectronic deep neural network, where the nonlinear response of the detector module can act as a hidden nonlinear layer. Also, the output of the ID₂N₂ can be considered as an optically pre-processed entry point to digital phase retrieval algorithms. The combination of ID₂N₂ with electronic networks will increase the performance of phase retrieval systems by leveraging the advantages of optical pre-processing with a diffractive network, such as the ability to discriminate the sign of a pupil phase, and reduce the requirements for computational resources and multiple sensors.’

- We added the results of our numerical experiments on the limited generalization ability of the network in Figure S8 and the following discussion at the end of the *ID₂N₂ numerical performance* section of the manuscript:

‘We evaluated the generalization ability of the proposed ID₂N₂ by training the network using two datasets consisting of Zernike polynomials (single Zernike polynomials Z₁ to Z₇ and single Zernike polynomials Z₈ to Z₁₄) that were each assigned a random RMS magnitude between ± 0.6 π and their corresponding complex field PSFs. Then, we tested the performance of the network in reconstructing single Zernike polynomials Z₁ to Z₁₄. The results reported in Figure S8 show that the networks trained with different subsets of Zernike polynomials have, to some extent, comparable generalization abilities, as confirmed by the average RMS errors calculated for the Zernike polynomials Z₁ to Z₁₄. However, the prediction ability drops notably when the network predicts polynomials that were not included in the training, as shown by the average RMS errors calculated on the subsets of polynomials.’

Second, the authors claim that “This compact device operates in the NIR wavelength region and processes complex field PSFs through all-optical inference, performing direct retrieval of the incoming field pupil phase, including the magnitude and the sign.”. But I don’t think their system is “all-optical” since they explicitly mention “optical-electronic conversion step” for nonlinear activation, which is not clear how to be characterized nor how efficient, while they mention 90Hz as their sensor refresh rate. It is very unclear how this nonlinear activation function is implemented. While the authors claim “The nonlinear activation function is implemented during the photoelectric conversion via the imaging sensor.” and model it with some ReLU function, it would be great to clarify which nonlinearity they are referring to: (is it just the photoelectric effect or some Gamma/square curves?)

Response:

We thank the Reviewer to raise a point that allows us to better explain our work.

The Reviewer states correctly that the system is not “all-optical”, since the optical-electronic conversion via the detector is part of the ID₂N₂ framework. To avoid confusion, we removed the statement ‘all-optical’ from our manuscript and we clarified that the DN₂ performs optical inference, while the entire system is optoelectronic in the last paragraph of the *Introduction* section of the manuscript.

The choice to use the ReLU function to approximate the nonlinearity of the detector is motivated by the response curve for a light sensitive detector module consisting of photoelectric conversion and electronic readout circuit, which can be divided into three parts: the dark area, the linear area and the saturation area, as shown by the figure below. The dark area of the response curve shows the detector modules response to very low light intensities; the output value of the detector module in this area is noisy and unpredictable. After the Noise Equivalent Exposure (NEE) point is reached, the output of the detector module becomes linear until a point called the Saturation Equivalent Exposure (SEE). At this point, increasing the light intensity results in a nonlinear increase in the detector module output. Our experiments happen before the SEE point and therefore the response function can be approximated by a ReLU function.

Figure: A typical response curve of a light sensitive detector module. Image adapted from <https://www.baslerweb.com/en/sales-support/knowledge-base/frequently-asked-questions/what-is-sensitivity-and-why-are-sensitivity-statements-often-misleading/14987/>.

During the training, we studied the relation between the brightness level of the output image and the system performance by tuning the parameter t of ReLU function (see Figure S1). To adjust the brightness of an image, the value of all pixels can be changed by a constant. This can be achieved:

- In post processing, by adding a positive constant to all the image pixel values to make the image brighter or subtracting a positive constant from all of the pixel values to make the image darker. This can be achieved by applying a shifted ReLU function as transfer function.
- During the detection step, by adjusting the camera exposure parameters and in this way changing the NEE point.

Given the low impact of the parameter t on this specific task, for this qualitative experimental demonstration, we fixed exposure and gain conditions without optimizing the parameter t .

To clarify this point we added:

- The following sentence to the first paragraph of the *Results* section of the manuscript: ‘The choice to use the ReLU function to approximate the nonlinearity of the detector is motivated by the response curve for a light sensitive detector module consisting of photoelectric conversion and electronic readout circuit (see Supplementary Information)’.
- The above discussion on the ReLU function to the *TensorFlow-based design and training* section of the Supporting Information.

My other concern is that the DNNs might not have the capability to retrieve arbitrary phases, due to their inherent linearity. Then “nonlinearity at last” scheme mentioned in the previous paragraph is actually more problematic since it makes it unclear how to increase the system’s capability although the authors set the scope of the work as “Our goal throughout this work was not to match the highest

accuracy achieved with a state-of-the-art ultra-deep CNN trained for phase retrieval tasks^{49,50}, but rather to understand and characterize the behavior of the ID₂N₂ when interacting with complex fields, proofing the ability to retrieve phase information and assess its potential in a hybrid optoelectronic architecture.”. I think the performance wouldn’t scale with the number of diffractive layers since they are all linear, which makes it challenging to retrieve the arbitrary (pupil) phase as they claim. It would be great if the authors could add analysis on the capacity of the system, or the scalability of its non-linearity.

Response:

As the Reviewer correctly observed, the DN₂s presented here consist of four diffractive layers of linear materials, and, without including the equivalent of a nonlinear activation function within the optical network, the performance of the system does not scale arbitrary with the number of diffractive layers.

Although the following papers presented results showing that multiple linear diffractive layers statistically perform better compared to a single layer trained for the same given task:

- X. Lin *et al.*, *All-optical machine learning using diffractive deep neural networks*. *Science*. 361, 1004–1008 (2018);
- Mengu, Y. *et al.*, *Analysis of diffractive optical neural networks and their integration with electronic neural networks*. *IEEE Journal of Selected Topics in Quantum Electronics* (2019) 26(1) 3700114;

the topic of the observed scalability in all-linear multilayer DN₂s and how this relates to the *depth* as understood in machine learning is a topic of ongoing scientific discussion. See the following papers:

- Wei, Haiqing, *et al.* "Comment on" *All-optical machine learning using diffractive deep neural networks*." *arXiv preprint arXiv:1809.08360* (2018).
- Mengu, Deniz, *et al.* "Response to Comment on" *All-optical machine learning using diffractive deep neural networks*." *arXiv preprint arXiv:1810.04384* (2018).

From our simulations (the results are reported in Figure S1), we empirically observed that, by applying a ReLU function as last layer of the system, it is possible to improve the performance of the network, especially for classification tasks (Figure S1c). However, we agree with the Reviewer that the application of the ReLU function as last layer of the network constitutes an architectural difference compared with computer-based neural networks and does not give the full benefit of depth and nonlinearity, as intended in machine learning. If we incorporate the optoelectronic neural network architecture presented here in a larger optoelectronic scheme, as in the approach proposed by Zhou, T., Lin, X., Wu, J. *et al.* Large-scale neuromorphic optoelectronic computing with a reconfigurable diffractive processing unit. *Nat. Photonics* 15, 367–373 (2021) (<https://doi.org/10.1038/s41566-021-00796-w>), the response of the detector module will play the role of the first nonlinear hidden layer in the larger network.

To clarify the concerns raised by the Reviewer:

- We added an analysis on the generalization capacity of the system in Figure S8 and in the last paragraph of section 2.2 of the manuscript:
‘We evaluated the generalization ability of the proposed ID₂N₂ by training the network using two datasets consisting of Zernike polynomials (single Zernike polynomials Z₁ to Z₇ and single Zernike polynomials Z₈ to Z₁₄) that were each assigned a random RMS magnitude between $\pm 0.6 \pi$ and their corresponding complex field PSFs. Then, we tested the performance of the network in reconstructing single Zernike polynomials Z₁ to Z₁₄. The results reported in Figure S8 show that the networks trained with different subsets of Zernike polynomials have, to some

extent, comparable generalization abilities, as confirmed by the average RMS errors calculated for the Zernike polynomials Z_1 to Z_{14} . However, the prediction ability drops notably when the network predicts polynomials that were not included in the training, as shown by the average RMS errors calculated on the subsets of polynomials.'

- We removed any reference to "arbitrary phase retrieval" from the manuscript.
- We added a discussion on the role of the ReLU function in the first paragraph of the *Discussion* section of the manuscript:

'In the optoelectronic network architecture presented here, the ReLU is applied as last output layer that adds a bias term to the output image – i.e. changes the value of all pixels by a constant, adjusting the brightness of the image – improving the performance of the network (**Figure S1b** and **Figure S1c**). The application of the nonlinear activation function as last layer of the network constitutes an architectural difference compared with computer-based neural networks. However, the ID₂N₂s could be incorporated in a more complex optoelectronic scheme, where the response of the detector module would play the role of the first nonlinear hidden layer of the network.'

- We discussed the potential of the proposed framework in a hybrid optoelectronic architecture in the *Conclusions* section of the manuscript:

'The ID₂N₂ presented here allows for direct retrieval of Zernike-based pupil phase, although the error rates may be too high for certain applications. To overcome this limitation, the ID₂N₂ can be used as a building block in larger scale optoelectronic deep neural network, where the nonlinear response of the detector module can act as a hidden nonlinear layer. Also, the output of the ID₂N₂ can be considered as an optically pre-processed entry point to digital phase retrieval algorithms. The combination of ID₂N₂ with electronic networks will increase the performance of phase retrieval systems by leveraging the advantages of optical pre-processing with a diffractive network, such as the ability to discriminate the sign of a pupil phase, and reduce the requirements for computational resources and multiple sensors.'

Additionally, the following might be confusing to readers:

- In the abstract, they claim an average RMS error of 0.036 pi radians but I guess this is in numerical simulation, not the experimental results. This should be clarified.

Response:

It is indeed the numerical result. To clarify this point we made the following changes:

- we removed the sentence 'with an average root mean square error of 0.036 π radians' from the *Abstract*
- we rephrase the following sentence in section 2.2 of the manuscript:
'It was numerically found that the average RMS error is 0.036 π radians'.

- It is confusing if this system can fully support 2pi range retrieval or those are scaled from 1.2pi range, since Fig. 2 shows 2pi range colorbars while the manuscript does mention 1.2pi range. I guess it is stretched somehow and the network itself has limited capability of supporting 2pi range though, this should be clarified.

Response:

We thank the Reviewer for pointing out this oversight.

The non degenerate response of the network to defocus, was evaluated using training and test data sets generated by each of the fourteen Zernike polynomials, where the random root-mean-square (RMS)

magnitude of each individual polynomial was varied between $\pm 1\pi$ radians. The PSF were calculated at the focal spot. Figure 2 reports the correct colorbars. The performance of the network in reconstructing single Zernike polynomial pupil phases was evaluated using training and test data sets generated by each of the fourteen Zernike polynomials, where the random root-mean-square (RMS) magnitude of each individual polynomial was varied between $\pm 0.6\pi$ radians.

- We corrected the colorbars of the pupil phases reported in Figure 3, Figure 4, Figure S3 and Figure S9
- We added a clarification in the *Methods, Training and test dataset processing* section of the manuscript:
‘In our numerical experiments we train an ID_2N_2 using a dataset consisting of 8000 randomly selected single Zernike polynomials between Z_1 and Z_{14} that were each assigned a random root-mean-square (RMS) magnitude between $\pm 0.6\pi$ and the corresponding complex field PSF.’
‘The non-degenerate response of the network to defocus, was evaluated using training and test data sets generated by each of the fourteen Zernike polynomials, where the random root-mean-square (RMS) magnitude of each individual polynomial was varied between $\pm 1\pi$ radians’.

- I do see strikethroughs on the optimised ReLU function, what are they?

Response:

The strikethroughs on the optimised ReLU function, like in the caption of Figure 5, are from a previous version of the manuscript. We removed them from the manuscript and apologize for our oversight.

- The TPN should be expanded when it is introduced first.

Response:

We corrected our oversight and explained the meaning of TPN (two-photon nanolithography) in *ID₂N₂ vectorial printing* section of the manuscript, when it was first introduced.

‘After calculating the phase delay of each diffractive neuron of the four diffractive layers comprising the DN_2 (**Figure 3a**), the 3D models of the DN_2 were obtained by converting the calculated phase value of each diffractive neuron into a relative height map and then printed in IP-S photoresist using a two-photon nanolithography (TPN) method’.

- In lines 391-399, “The performance of the network in reconstructing combinations of Zernike polynomial pupil phases was evaluated using a test data set comprising 8000 PSFs with pupil phases generated using a linear combination of Zernike polynomials from Z_1 to Z_{14} . The magnitude of each individual pupil phase was varied between $\pm 1\pi$ radians to study the response to different aberration magnitudes” This paragraph is just repeated with slightly different numbers.

Response:

We thank the Reviewer for the comment that allows us to clarify our presentation.

In our numerical experiments we train an ID_2N_2 using a dataset consisting of 8000 randomly selected single Zernike polynomials between Z_1 and Z_{14} that were each assigned a random root-mean-square (RMS) magnitude between $\pm 0.6\pi$ and the corresponding complex field PSF. We then test the ability of the same network to reconstruct:

- **single Zernike polynomial** pupil phases using a test data set comprising 800 PSFs with pupil phases generated by each of the fourteen Zernike polynomials used in the training data set. For

this test, the random root-mean-square (RMS) magnitude of each individual polynomial was varied between $\pm 0.6\pi$ radians such that the entire training range was probed.

- **combinations of Zernike polynomial** pupil phases using a test data set comprising 8000 PSFs with pupil phases generated using linear combination of Zernike polynomials from Z_1 to Z_{14} . The magnitude of each individual pupil phases was varied between $\pm 1\pi$ radians and the combinations were then normalized to $\pm 0.2\pi$, $\pm 0.4\pi$, $\pm 0.6\pi$, $\pm 0.8\pi$ and $\pm 1\pi$ radians to study the response to different aberration magnitudes.

We rephrase the last paragraph of the *Training and test dataset processing* section to clarify this concept:

'In our numerical experiments we train an ID_2N_2 using a dataset consisting of 8000 randomly selected single Zernike polynomials between Z_1 and Z_{14} that were each assigned a random root-mean-square (RMS) magnitude between $\pm 0.6\pi$ and the corresponding complex field PSF. We then test the ability of the same network to reconstruct:

- *single Zernike polynomial* pupil phases (Figure 3c) using a test data set comprising 8000 PSFs with pupil phases generated by each of the fourteen Zernike polynomials used in the training data set. For this test, the random root-mean-square (RMS) magnitude of each individual polynomial was varied between $\pm 0.6\pi$ radians such that the entire training range was probed.
- *combinations of Zernike polynomial* pupil phases (Figure 3e) using a test data set comprising 8000 PSFs with pupil phases generated using linear combination of Zernike polynomials from Z_1 to Z_{14} . The magnitude of each individual pupil phases was varied between $\pm 1\pi$ radians and the combinations were then normalized to $\pm 0.2\pi$, $\pm 0.4\pi$, $\pm 0.6\pi$, $\pm 0.8\pi$ and $\pm 1\pi$ radians to study the response to different aberration magnitudes.'

Reviewer #3 (Remarks to the Author):

In this paper, Goi et al demonstrated a 4-layer diffractive neural network fabricated by two-photon polymerization that can convert intensity point spread function to intensity distribution that is proportional to the pupil phase that resulted in the point spread function. I agree with the authors that real-time analysis of optical wavefront is a promising application of diffractive neural networks, and the paper took a systematic approach with well executed experiments to prove the practicality of this method for NIR wavelengths. However, there are several key technical aspects that need to be clarified in order to enable readers to truly evaluate the potential of the method:

Response:

We thank the Reviewer for the time dedicated to review our work and the insightful comments she/he provided. We hope to be able to address her/his concerns in our response.

1. What is the role of ReLU layer in the CCD or CMOS camera? Since nonlinear layers only serve to improve the performance of neural networks when they function as hidden layers instead of the last output layer and the authors claim that no more digital operation was needed after reading out the light intensity distribution from the camera, it is not very clear how a shifted (or so-called optimised) ReLU function improves the performance other than adding a bias term to the image. The claim of ReLU nonlinear function can be potentially misleading for the readers to believe the nonlinear operation contributes to the computation as those in deep neural networks, which is not true in this study.

Response:

We thank the Reviewer for this question that allows us to clarify our work.

The Reviewer states correctly that, in our optical network architecture, the ReLU is applied as last output layer, and, when the parameter $t > 0$, the ReLU function adds indeed a bias term to the output image – i.e. changes the value of all pixels by a constant, adjusting the brightness of the image. During the training, we studied the relation between the brightness level of the output image and the system performance by tuning the parameter t of ReLU function (see Figure S1). To adjust the brightness of an image, the value of all pixels can be changed by a constant. This can be achieved:

- In post processing, by adding a positive constant to all the image pixel values to make the image brighter or subtracting a positive constant from all of the pixel values to make the image darker. This can be achieved by applying a shifted ReLU function as transfer function.
- During the detection step, by adjusting the camera exposure parameters.

From our simulations (Figure S1,) we empirically observed that, by tuning the parameter t (shifting the ReLU function), it is possible to improve the performance of the network, especially for classification tasks (Figure S1c). We believe that this happens because, in the case of classification tasks, the goal of a diffractive neural network is to focus all the light in output on a specific spot associated to the class of the input image (Figure 1b of *X. Lin et al., All-optical machine learning using diffractive deep neural networks. Science. 361, 1004–1008 (2018)*), and changing the bias of the detector module filters out noise, making it easier to determine the class of the input image. In the application presented in this work, phase retrieval, we agree with the Reviewer that the role of the of the ReLU function is marginal, as also empirically investigated in Figure S1b.

As the Reviewer stated, the application of the ReLU function as last layer of the network constitutes an architectural difference compared with computer-based neural networks and does not give the full

benefit of depth and nonlinearity, as intended in machine learning (as discussed in detail in the response to Reviewer #2 question 3). If we incorporate the optoelectronic neural network architecture presented here in a larger optoelectronic scheme, as in the approach proposed by Zhou, T., Lin, X., Wu, J. *et al.* Large-scale neuromorphic optoelectronic computing with a reconfigurable diffractive processing unit. *Nat. Photonics* 15, 367–373 (2021) (<https://doi.org/10.1038/s41566-021-00796-w>), the response of the detector module will play the role of the first nonlinear hidden layer in the larger network.

To clarify the points discussed above we:

- reparsed the last paragraph of the *Introduction* of the manuscript:
‘The resulting intensity distribution is then detected by the sensor, thus adding the nonlinear responsivity required to achieve an integrated diffractive deep neural network (ID₂N₂) module with phase retrieval capabilities. We show numerically and experimentally that these optoelectronic networks can retrieve the pupil phase of incoming PSFs with low error rates and, through co-integration with standard complementary metal oxide semiconductor (CMOS) imaging sensors, have the potential for leading to a new generation of compact optoelectronic wavefront sensors.’
- added a discussion on the role of the ReLU function in the first paragraph of the *Discussion* section of the manuscript:
‘In the optoelectronic network architecture presented here, the ReLU is applied as last output layer that adds a bias term to the output image – i.e. changes the value of all pixels by a constant, adjusting the brightness of the image – improving the performance of the network (**Figure S1b** and **Figure S1c**). The application of the nonlinear activation function as last layer of the network constitutes an architectural difference compared with computer-based neural networks. However, the ID₂N₂s could be incorporated in a more complex optoelectronic scheme, where the response of the detector module would play the role of the first nonlinear hidden layer of the network.’

2. Is the mapping from the pupil phase to intensity PSF a linear one? if so, the author should show the derivation. If not, how could the spatial modes be transformed by a diffractive neural network, which performs linear operations? if the authors believe the diffractive neural network performs nonlinear operation to some capacity, say with quadratic nonlinearity from amplitude to intensity detection, the authors should also carefully discuss this point. I say “to some capacity” since the quadratic nonlinearity, even as hidden layers, does not make the diffractive neural network a universal approximator, so it does not give the full benefit of nonlinearity to the neural networks. The authors are welcome to make arguments on this point.

Response:

To numerically train the ID₂N₂s we did not use the PSF intensities, but we used the complex field of the PSFs, which are linear expressions of the pupil phases. The detailed derivation of the complex field calculations that we perform to create our training and test datasets is described in the paper by Leutenegger, M., et al., Fast focus field calculations, *Opt. Express* 14, 11277-11291 (2006) (<https://doi.org/10.1364/OE.14.011277>) that we cited in reference 24 of the original manuscript.

In the task presented in this work, the DN₂ maps a complex field received in input into another complex field in output, in a linear operation. The sensor detects the intensity of the output field.

The DN₂ presented here consist of four diffractive layers based on linear materials. Without including the equivalent of a nonlinear activation function within the optical network, we agree with the Reviewer that the application of the ReLU function as last layer, does not give the full benefit of depth and nonlinearity, as intended in machine learning, therefore, the ID₂N₂s presented here are not universal approximators.

To clarify the points discussed above we:

- Repaired the first paragraph of the *Results* section of the manuscript:
‘The DN₂ receives in input a complex field representing a PSF, which is a linear expression of the pupil phase distortion in form of a single Zernike polynomial or a superposition of multiple superimposed Zernike polynomials. Retaining phase and amplitude information of the incoming light, the DN₂ is able to scatter and modulate each of a multitude of aberrated PSFs, mapping them into a specific output intensity pattern that shows the original pupil phase in magnitude and sign, in a linear operation. The complex field PSFs used during training were calculated following a fast Fourier transform (FFT) implementation of vectorial Debye theory, while the complex field PSFs in the experiment were generated by the optical setup reported in Figure 4d.’
- Added a discussion on the role of the ReLU function in the first paragraph of the *Discussion* section of the manuscript:
‘The application of the nonlinear activation function as last layer of the network constitutes an architectural difference compared with computer-based neural networks. Without including the equivalent of a nonlinear activation function within the diffractive layers of the optical network, the ID₂N₂ can perform only linear operations. However, the nonlinear layer could act as a hidden layer in the deep network if incorporated in a larger scheme or in loop.’

3. Would the authors elaborate more on the following statement? “It is important to note that an ID₂N₂ trained using only the intensity pattern of the PSFs is not able to distinguish the sign of defocus, while the ID₂N₂ trained using the complex fields that describe the PSFs can retrieve the information contained within the phase structure of the PSF and encode it in the intensity of the outputs 24 ” Does it mean the intensity measured at plane 1 in Figure 4d and e cannot be used directly for training the diffractive neural network? Instead, either the phase of the PSF also needs to be measured at plane 1 or a phase pattern needs to be assumed for each measured PSF in order for the training to work well?

Response:

We thank the Reviewer for this question that allows us to better explain our work.

To numerically train the ID₂N₂s to retrieve the phase of the PSF we did not use the PSF intensities, but we used the complex field of the PSFs. The phase terms/complex fields at the input plane used for training of the DN₂s was neither measured nor assumed, but rather calculated using the Fast Fourier Field method as outlined in reference 24 (Leutenegger, M., et al., Fast focus field calculations. **14**, 4897–4903 (2006)). The training using complex fields allows the network to have non-degenerate response to symmetric aberrations, such as defocus. For example, if we apply defocus (Z_4) with same amplitude but opposite sign to the incoming beam, the PSFs will have the same intensity distributions (Figure 2a), however, the real and imaginary input fields will differ for the two cases. If the network is trained with the intensities of the PSFs, the inputs in case of defocus with same intensity and opposite sign will be identical and the network will not be able to map them into different outputs (Figure 2b). To have an ID₂N₂ able to discriminate symmetric aberrations, such as defocus, it is necessary to perform the training using the complex field PSFs (Figure 2c).

In Figure 4e we compared the numerically calculated PSF intensities with the experimentally detected PSF intensities to validate our fast focus field calculations. The images of the experimental PSF intensities reported in Figure 4e are not used during training.

We clarified this point:

- Reparsing the first paragraph of the Results section:
'The DN_2 receives in input a complex field representing a PSF, which is a linear expression of the pupil phase distortion in form of a single Zernike polynomial or a superposition of multiple superimposed Zernike polynomials. Retaining phase and amplitude information of the incoming light, the DN_2 is able to scatter and modulate each of a multitude of aberrated PSFs, mapping them into a specific output intensity pattern that shows the original pupil phase in magnitude and sign, in a linear operation. The complex field PSFs used during training were calculated following a fast Fourier transform (FFT) implementation of vectorial Debye theory, while the complex field PSFs in the experiment were generated by the optical setup reported in Figure 4d.'
- Adding a sentence in the *Training and test dataset processing* section:
'The numerical calculations of the PSF were validated by the comparison between the numerically calculated PSF intensities and the experimentally detected PSF intensities reported in Figure 4e and Figure S9.'

4. What is the real generalization ability of the diffractive neural network for Zernike polynomials? Even though the study used differently randomly generated linear combination of Zernike polynomials for train and test respectively, they were still trained on the same basis of Zernike polynomials.

Response:

We want to take the opportunity to clarify training and testing methods: the network presented in the main text was trained using only single Zernike polynomials, each assigned a random root-mean-square (RMS) magnitude between $\pm 0.6 \pi$ and the corresponding complex field PSFs. The network's ability to retrieve pupil phases was then tested using single Zernike polynomials and differently randomly generated linear combination of Zernike polynomials, which already shows some generalization ability. In the Supplementary Information, we also evaluated the performance of an ID_2N_2 trained on a data set comprising pupil phases and the corresponding PSFs generated from combinations of Zernike polynomials in reconstructing the pupil phase generated from single and combinations of Zernike polynomials. The results are reported in **Figure S6** and **Figure S7**.

Essentially, I wish to know, for example, if the train set contains Z1 to Z7, and the test set contain Z8-Z14, would the training result generalize to test? In a more fundamental way, the question is about whether the linear operation of the diffractive neural network is truly able to model a physical system of such complexity, and whether it will become much more accurate with further developments.

Response:

The Reviewer raised an interesting point.

We tested the generalization ability of the proposed ID_2N_2S , with numerical experiments. We trained the ID_2N_2S using two datasets:

- a dataset consisting of 4000 randomly selected single Zernike polynomials between Z_1 and Z_7 that were each assigned a random root-mean-square (RMS) magnitude between $\pm 0.6 \pi$ and the corresponding complex field PSF.
- a dataset consisting of 4000 randomly selected single Zernike polynomials between Z_8 and Z_{14} that were each assigned a random root-mean-square (RMS) magnitude between $\pm 0.6 \pi$ and the corresponding complex field PSF.

Then, we evaluated the generalization ability of the two networks by testing the performance when reconstructing single Zernike polynomials Z_1 to Z_{14} . The results are reported in the following figure:

The ID₂N₂s were trained under the same conditions as the network reported in Figure 3. The results

show that the networks trained with different subsets of Zernike polynomials, have, to some extent, comparable generalization abilities, as confirmed by the average RMS errors calculated for the Zernike polynomials Z_1 to Z_{14} . However, the prediction ability drops notably when the network predicts polynomials that were not included in the training, as shown by the average RMS errors calculated on the subsets of polynomials.

The ID₂N₂ presented here allows for direct retrieval of Zernike-based pupil phase, although the error rates may be too high for certain applications. To overcome this limitation, the ID₂N₂ can be used as a building block in larger scale optoelectronic deep neural network, where the nonlinear response of the detector module can act as a hidden nonlinear layer. Also, the output of the ID₂N₂ can be considered as an optically pre-processed entry point to computer-based digital phase retrieval algorithms. The combination of ID₂N₂ with electronic networks will increase the performance of phase retrieval systems by leveraging the advantages of optical pre-processing with a diffractive network, such as the ability to discriminate the sign of a pupil phase, and reduce the requirements for computational resources and multiple sensors

To discuss the points raised by the Reviewer we performed the following changes to the manuscript:

- We added the results of our numerical experiments on the generalization ability of the network in Figure S8 and the following discussion at the end of the section 2.2 of the manuscript:
‘We evaluated the generalization ability of the proposed ID₂N₂ by training the network using two datasets consisting of Zernike polynomials (single Zernike polynomials Z₁ to Z₇ and single Zernike polynomials Z₈ to Z₁₄) that were each assigned a random RMS magnitude between $\pm 0.6 \pi$ and their corresponding complex field PSFs. Then, we tested the performance of the network in reconstructing single Zernike polynomials Z₁ to Z₁₄. The results reported in Figure S8 show that the networks trained with different subsets of Zernike polynomials have, to some extent, comparable generalization abilities, as confirmed by the average RMS errors calculated for the Zernike polynomials Z₁ to Z₁₄. However, the prediction ability drops notably when the network predicts polynomials that were not included in the training, as shown by the average RMS errors calculated on the subsets of polynomials.’
- We added the following discussion on the applicability of the method in the *Conclusion* section of the manuscript:
‘The ID₂N₂ presented here allows for direct retrieval of Zernike-based pupil phase, although the error rates may be too high for certain applications. To overcome this limitation, the ID₂N₂ can be used as a building block in larger scale optoelectronic deep neural network, where the nonlinear response of the detector module can act as a hidden nonlinear layer. Also, the output of the ID₂N₂ can be considered as an optically pre-processed entry point to digital phase retrieval algorithms. The combination of ID₂N₂ with electronic networks will increase the performance of phase retrieval systems by leveraging the advantages of optical pre-processing with a diffractive network, such as the ability to discriminate the sign of a pupil phase, and reduce the requirements for computational resources and multiple sensors.’

5. My understanding is that the authors assumed a uniform amplitude distribution at the pupil plane, and then only seek to recover phase pattern at the pupil plane. What if there is actually an intensity distribution at the pupil plane, is it possible to infer the amplitude AND phase of the field based on the intensity PSF? Have the authors performed simulation on this? Please comment on it.

Response:

The Reviewer states correctly that in the simulations that we performed to create the training datasets, we just considered a point source. More in detail, we assumed a coherent, monochromatic wave field with uniform amplitude distribution parallel to the optical axis of the simulated objective crossing the aperture stop (entrance pupil), and the DNNs in this work are trained to recover only phase patterns.

The point raised by the Reviewer is interesting in context of confocal multifocal or STED microscopy and in direct laser writing. A non-uniform intensity distribution can be incorporated in the derivations of the PSFs, described in Leutenegger, et al., "Fast focus field calculations," Opt. Express 14, 11277-11291 (2006) (<https://doi.org/10.1364/OE.14.011277>).

In the current configuration, we are projecting a 2D input (uniform amplitude and non-uniform phase) on a 1D output (phase encoded in an intensity map) and in doing so we are able to retrieve the phase information including the sign of symmetric aberrations, such as defocus. If we include variation in amplitude across the pupil phase, we are in the situation of projecting a 2D input (non-uniform amplitude and non-uniform phase) on a 2D output (phase and amplitude encoded in an intensity map). Our hypothesis is that the system in the current configuration will not work properly and adjustments

should be made to allow the retrieval of all the information – e.g. the introduction of a second detector. Testing this hypothesis requires a set of experiments that constitute an interesting outlook of this work.

We added the following comment in the main text, *Training and test dataset processing* section to clarify the framework of our experiments:

‘A coherent, monochromatic wave field with uniform amplitude distribution parallel to the optical axis of the simulated objective crossing the aperture stop (entrance pupil) was considered in the current experiments.’

6. The authors should discuss the difference and similarities of their method from spatial mode sorting with multiple diffractive elements, for example, demonstrated in the following paper with multi-plane light conversion: <https://doi.org/10.1038/s41467-019-09840-4> It is not very clear to me how the demonstrated experiment is more than a mode sorter that maps a linear combination of a set of spatial modes to a linear combination of another set of modes with the same linear coefficients.

Response:

We thank the Reviewer for raising this point that gives us the opportunity to clarify our presentation.

If we consider the method that we proposed and a conventional spatial mode sorter, we have that in both cases, a complex input, a beam that consist of many orthogonal spatial components, is transformed by the system into an output that gives information on the input mode composition.

However, the approaches followed by the Laguerre-Gaussian (LG) mode sorter and by our ID₂N₂ are different. A conventional spatial mode sorter, like the Laguerre-Gaussian (LG) mode sorter demonstrated by Fontaine et al., maps each spatial component of the input beam into its own independent Gaussian spot in a grid on the output plane. This can be seen as a physical implementation of a modal-based approach to phase retrieval. On the other hand, the method presented in this work, is a physical implementation of a direct phase retrieval scheme. The ID₂N₂s map a light beam into an intensity distribution representing the beam’s pupil phase, an image that gives information on the beam’s original aberration.

The ID₂N₂ can be trained to perform sorting, multiplexing and de-multiplexing of light beams, following a modal-based approach. For example, in the work Zhang et al., Polarized deep diffractive neural network for sorting, generation, multiplexing, and de-multiplexing of OAM modes, *Opt. Express* 30, 26728-26741 (2022)) the authors propose a system able to sort 14 kinds of polarized vortex beams with different topological charges into 14 Gauss beams in 14 different circle areas of the detecting plane, achieving the performance of a conventional mode sorter.

We discussed the point raised by the Reviewer in the *Discussion* section of the manuscript:

Other than a spatial mode sorter that implements a modal-based approach to phase retrieval, our method is a physical implementation of a direct phase retrieval scheme, where the ID₂N₂s map a light beam into an intensity distribution representing the beam’s Zernike-based pupil phase, an image that gives information on the beam’s original aberration. However, the ID₂N₂ can be trained to perform spatial mode sorting, multiplexing and de-multiplexing of light beams, following a modal-based approach.’

Other technical issues:

1. Why does the phase plate appear to be axially symmetric in Figure 2a? Were they trained exclusively with central symmetric Zernike patterns?

Response:

The phase plates in Figure 3a are the result of training with a dataset obtained generating pupil phase functions using single Zernike polynomials from Z_1 to Z_{14} (OSA/ANSI indices) with random intensity, as detailed in the second paragraph of page 8 in the main text, *Methods* and *Supporting Information*.

The plates present some degree of radial symmetry (especially in layers 1 and 2) and axial symmetry (especially layer 3 and 4) that reflect the symmetries in the Zernike polynomials used during the training.

We would like to use this opportunity to clarify that the diffractive layers shown in the original Figure 3a were not the ones of the ID_2N_2 discussed in the main manuscript. We corrected our oversight in the revised manuscript.

2. I am curious why the diffractive neural network output trained with complex field in Figure 2c has more fine structures compared to Figure 2b trained with intensity field. Are those feature or artifacts?

Response:

The DNNs from Figure 2b and 2c are trained under the same conditions and with the same number of PSFs. However, the network in Figure 2b was trained given in input the PSF intensities while the network in Figure 2c was trained using the complex field PSFs. As shown in Figure 2a, the PSF intensities for pupil phases with same amplitude and opposite sign are the same.

Our hypothesis is that a basic DNN trained to map identical intensity inputs into different output patterns, like shown in Figure 2c, when it encounters an ambiguous input, predicts an output that is an average of the multiple outputs given during the training (see the following discussion: <https://stats.stackexchange.com/questions/96225/neural-network-what-if-there-are-multiple-right-answers-for-a-given-set-of-inpu>). This hypothesis is supported by the smoother and uniform outputs reported in Figure 2b, however, the problem should be further investigated to confirm our hypothesis.

To clarify this point we added the following sentence to section 2.1 of the manuscript:

‘The outputs of ID_2N_2 trained using only the intensity patterns are smoother and more uniform compared to the outputs of a ID_2N_2 trained using the complex fields, since the network appears to map the ambiguous inputs in an image that is the average of the multiple outputs given during the training.’

3. There seems to be inconsistency on which camera is used for detection and implements nonlinearity. CMOS camera is mentioned in section 2.5 and Figure 4, but CCD camera is mentioned everywhere else for nonlinear ReLU functions. Is the nonlinear function really caused by “photoelectric conversion” as claimed by the authors, or just some analog circuit setting? Regardless of these details, I respectfully disagree with the claim that ReLU nonlinear functions play any significant role in this experiment.

Response:

The detector used in all the experimental characterization of the ID₂N₂ is a CCD camera. Therefore, in the *Experimental performance* and in the *Methods, Characterization setup* sections, we refer to the CCD camera.

We printed an ID₂N₂ on a CMOS detector to demonstrate the integrability of these optical networks on commercial CMOS chips. Therefore, we mentioned the CMOS in the introduction, the *Four-layer ID₂N₂ printed on CMOS* section, the *Conclusions* and in the *Methods, Nanoprinting* sections of the manuscript.

To clarify this point, we added the following sentence in the *Discussion* section of the manuscript:

‘The experimental characterization of the ID₂N₂ was performed using a CCD camera, while the integration of DN₂ with a CMOS detector module aimed to prove the integrability of our optoelectronic framework. The size of the diffractive elements presented here (30×30 μm²) corresponds to 27×27 pixels on a Sony IMX219PQ CMOS image sensor with pixel size 1.12×1.12 μm², these images would be too low in resolution to appreciate the functionality of the optical network.’

The choice to use the ReLU function to approximate the nonlinearity of the detector is motivated by the response curve for a light sensitive detector module consisting of photoelectric conversion and electronic readout circuit, as shown by the figure below.

A typical response curve of a light sensitive detector module. Image adapted from <https://www.baslerweb.com/en/sales-support/knowledge-base/frequently-asked-questions/what-is-sensitivity-and-why-are-sensitivity-statements-often-misleading/14987/>.

Regarding the role of the of the nonlinear ReLU function in this specific application of the ID₂N₂, the Reviewer states correctly that is marginal, as shown by Figure S1b. However, we choose to include and discuss the role of the ReLU function in our framework because, for other tasks, such as classification, it has a strong impact on the system performance (Figure S1c).

The physical basis for choosing the ReLU function were discussed in detail in the replies to Reviewer #1, question 1, Reviewer #2, question 2 and Reviewer #3, question 2. To clarify this point we added:

- The following sentence to the first paragraph of the Results section:
‘The choice to use the ReLU function to approximate the nonlinearity of the detector is motivated by the response curve for a light sensitive detector module consisting of photoelectric conversion and electronic readout circuit (see Supplementary Information)’.

- The discussion above on the ReLU function to the *TensorFlow-based design and training* section in the Supporting Information.

4. The following statements are confusing:

“Each neuron is printed in a single step as a thin rod that extends into the axial direction, allowing for precise control of the length – and hence bias – of each printed artificial neuron.”

“While the neurons of the input and output layers are unbiased (i.e., uniform), each neuron of a diffractive layer adds a bias in the form of a phase delay to the transmitted signal.”

This is because it is a convention in machine learning community that the bias of artificial neurons refer to the b term in the update equation of $Wx+b$. However, what the ‘bias’ actually means here is the phase bias, which has a completely different meaning. Since there is really no one-to-one analogy between the mathematical expression of diffractive neural network and digital neural networks (see Figure 1D of <https://doi.org/10.1126/science.aat8084>), using the same term (e.g., artificial neurons, bias) to address different mathematical entities in the two models only serve to confuse readers. The authors should try to make clear and unambiguous statements.

Response:

The Reviewer raised a good point. In our manuscript we adopted the terminology from the seminal work of Lin et al. (<https://doi.org/10.1126/science.aat8084>), cited also by the Reviewer, that defined these systems as ‘*coherent diffractive network modelled by physical wave propagation to connect various layers through the phase and amplitude of interfering waves, controlled with multiplicative bias terms and physical distances*’ (see Supplementary Material, Comparison with standard neural networks section) and the pixels as ‘*neurons*’.

We agree with the Reviewer that using the terms ‘bias’ and ‘neurons’ without discussing the architectural differences with standard neural networks can lead to ambiguity. To avoid this ambiguity, we refer to DN₂ single pixels as ‘diffractive neurons’ and we replaced ‘bias’ with ‘phase delay’ throughout the revised manuscript. Moreover, in the Supporting Information, we revised the following sentences:

‘Each neuron of the diffractive layer adds a phase delay to the transmitted signal to map each input key into a specific output pattern $y_{out}(x, y, z)$. The phase delay of each optical neuron is adjusted during the training.’

5. I strongly encourage the authors to share their simulation code by depositing in a publicly available online repository.

Response:

The instructions to develop the code for the creation of the training and test datasets can be found in Leutenegger, et al., "Fast focus field calculations," Opt. Express 14, 11277-11291 (2006) (<https://doi.org/10.1364/OE.14.011277>)

The instructions to develop the base code for the training of the diffractive neural networks can be found in X. Lin et al., All-optical machine learning using diffractive deep neural networks. Science. 361, 1004–1008 (2018) (<https://doi.org/10.1126/science.aat8084>).

The deep learning models reported in this work used standard libraries and scripts that are publicly available in PyTorch. The scripts based on these publicly available libraries and published information are currently not in a stage to be published, but we are happy to share it on request.

We updated the *Code availability* statement to clarify this point:

‘The custom code and mathematical algorithm used to obtain the results within this paper may be requested from the corresponding author, E. G..’

6. My understanding is that a coherent laser source was used for this experiment. The authors should clearly state the demonstration was made with coherent light, and it is preferred to have a discussion on what would happen if the light source is incoherent.

Response:

We agree with the Reviewer that this point has not been properly explained in the manuscript. We added the missing information about the laser source in the *Experimental performance* and *Methods* sections.

- To demonstrate the ability of the ID₂N₂ to retrieve the phase of PSFs and to map it into an intensity distribution, we designed an experimental setup, which, using a coherent laser source and a spatial light modulator (SLM), allowed the projection of PSFs with arbitrary pupil phases at the input plane of the DN₂, and the detection of the output with a CCD camera.
- A schematic diagram of the experimental setup is given in Figure 4b. The light beam is generated through a Thorlabs OBIS 785 nm coherent laser source.

The DNN presented in this work are trained to operate with temporally and spatially coherent light at a 785 nm wavelength. The output of these networks will be unpredictable if the incoming light is not coherent. The reconstruction of phase distortion under spatially and/or temporally incoherent illumination is indeed a challenging task and an interesting outlook of this work. In this context, we can find the works of Luo *et al.* ‘Design of task-specific optical systems using broadband diffractive neural networks’ *Light Sci Appl* 8, 112 (2019) (<https://doi.org/10.1038/s41377-019-0223-1>), and Jiao *et al.*, ‘Optical machine learning with incoherent light and a single-pixel detector’ *Opt. Lett.* 44, 5186-5189 (2019) (<https://doi.org/10.1364/OL.44.005186>), exploring broadband DNNs architectures.

We added a discussion on the consequences of using an incoherent light source in the *Discussion* section of the manuscript:

‘The ID₂N₂ presented in this work are trained to operate with temporally and spatially coherent light at a 785 nm wavelength. The use of other wavelengths or spatially and temporally incoherent illumination^{38,50} on the networks presented in this work, will result in unpredictable outputs.’

Minor points:

1. “Matrixes” should be “matrices”.
2. Citation 1 and 2 are the same 3.
3. The author list of citation 30 seems to be wrong.

Response:

We thank the Reviewer for pointing out these oversights. We corrected 1. and 2. in the main text, and we rectified the list of authors of citation 30:

Chang, J., Sitzmann, V., Dun, X., Heidrich, W. and Wetzstein, G., Hybrid optical-electronic convolutional neural networks with optimized diffractive optics for image classification. *Sci Rep* 8, 12324 (2018). <https://doi.org/10.1038/s41598-018-30619-y>

REVIEWER COMMENTS

Reviewer #1 (Remarks to the Author):

The authors have adequately addressed my previous comments.

Reviewer #2 (Remarks to the Author):

The authors' answers on the capacity and scalability of this framework have not fully addressed my question, and they rather suggest using their prototype as a submodule of reconfigurable diffractive processing unit by Zhou et al. 2019, while their prototype is fixed and not reconfigurable. Also, the authors pointed out the number of pixels as the main reason for not demonstrating the functionality of the CMOS prototype, but this is not fully convincing to me because I guess they can create another dataset with small Zernike images. These still make me hesitate to recommend this paper for publication in Nature Comms.

I appreciate the thoughtful responses by the authors, and the manuscript has been significantly improved in clarity. In principle, I support the publication of the work in *Nature communications*.

While most of my concerns have been satisfyingly addressed, there are several remaining details I wish the authors can revise on.

1. The authors have pointed out that, according to Ref. 45 (Leutenegger, M., et al., Fast focus field calculations), the transformation from the complex-value E field of the PSF (E_{PSF}) to the complex-value E field at the aperture ($E_{aperture}$) is a linear one. However, at the output of the diffractive neural network, what is measured is the *intensity* $|E_{aperture}|^2$ instead of the complex-value E field, and what is intended to be measured at the output of the diffractive neural network is the *phase* of the complex-valued E-field $\arg(E_{aperture})$. First, these two quantities are nominally different, and I assume only under some conditions they are approximately equal. Second, it is not immediately obvious the linear relation between E_{PSF} and $E_{aperture}$ translates to the linear relation between E_{PSF} and $|E_{aperture}|^2$ (or $\arg(E_{aperture})$, if they are proven to be equal under some conditions). I hope the authors can clarify this point more.
2. In results section, the authors make the following statement: “While the nonlinear activation in the optical characterization of the printed DN₂s is implemented through the photoelectric conversion of the field incident on the CCD sensor³⁹, this behavior is approximated as a Rectified Linear Unit (ReLU) function of optimized shape (see **Figure S1** for optimization) for purposes of in-silico training and numerical characterization of the DN₂s. The choice to use the ReLU function to approximate the nonlinearity of the detector is motivated by the response curve for a light sensitive detector module consisting of photoelectric conversion and electronic readout circuit⁴⁶ (see Supplementary Information).” I suggest authors making an explicit statement here like “While the diffractive neural network performs a linear operation on the light field, the nonlinear response of the CCD sensor at the output of the diffractive neural network may serve as nonlinear activation function of a hidden layer, if the readout of the sensor is used as the input to another digital or optical neural network.” This is because the role of the nonlinear response of the CMOS sensor is really determined by the subsequent operations, whether it functions as a nonlinear activation in a hidden layer or not is decided by whether there are more layers after it. If it is the last layer, the nonlinear response of the camera only serves as a bias term, as we have discussed. Also, the ReLU-like nonlinear response is not the only way to implement nonlinear function with camera sensors, in fact, the intensity detection is a more basic one and can serve as a quadratic

nonlinearity, as mentioned in several previous works (Hamerly, Ryan, et al. "Large-scale optical neural networks based on photoelectric multiplication." *Physical Review X* 9.2 (2019): 021032. Also, Spall, James, Xianxin Guo, and Alexander I. Lvovsky. "Hybrid training of optical neural networks." *arXiv preprint arXiv:2203.11207* (2022).) Again, I believe the impact of this work does not hinge on whether there is a nonlinear activation implemented by the camera sensor or not, but on the application of diffractive neural network for direct phase retrieval.

The following response to the Reviewers comments for our manuscript “Direct retrieval of Zernike-based pupil functions using integrated diffractive deep neural networks” (NCOMMS-22-23143) is color-coded as follows:

- the original comments of the Reviewers are written in black
- our responses to the Reviewers comments are written in blue
- the changes to the manuscript are written in green

All the changes in the manuscript text file are highlighted in yellow.

REVIEWER COMMENTS

Reviewer #2 (Remarks to the Author):

The authors’ answers on the capacity and scalability of this framework have not fully addressed my question, and they rather suggest using their prototype as a submodule of reconfigurable diffractive processing unit by Zhou et al. 2019, while their prototype is fixed and not reconfigurable.

Response:

We thank the Reviewer for the time dedicated to review our work and hope to be able to address the Reviewer’s concerns in our response.

The Reviewer states correctly that, unlike the diffractive processing unit proposed by Zhou et al., *Nat. Photonics* 15, 367–373 (2021) (<https://doi.org/10.1038/s41566-021-00796-w>), our system is not reconfigurable. Therefore, the reference that we choose as an example of the optoelectronic network where our module can find application was misleading.

Our optoelectronic module (fixed diffractive elements and sensor) and an electronic deep neural network, or another differentiable image processing algorithm, can be jointly optimized for a specific task, then used to process an image optically and electronically. Together, our module and the electronic network form a hybrid optical–electronic neural network that can be used to perform inference tasks, such as classifying captured images more robustly, faster or using less power than conventional digital layers. The idea of optical encoder–electronic decoder systems in the context of deep optics has been discussed in the following works:

- Sitzmann, V. et al. End-to-end optimization of optics and image processing for achromatic extended depth of field and super-resolution imaging. *ACM Trans. Graph.* 37, 114 (2018).
- Martel, J. N. P., Muller, L. K., Carey, S., Dudek, P. & Wetzstein, G. Neural sensors: learning pixel exposures for HDR imaging and video compressive sensing with programmable sensors. *IEEE Trans. Pattern Anal. Mach. Intell.* 42, 1642–1653 (2020).
- Wetzstein, G. et al. Inference in artificial intelligence with deep optics and photonics. *Nature* 588, 39–47 (2020).

To clarify this point:

- We remove all the references to Zhou et al., *Nat. Photonics* 15, 367–373 (2021) in relation to the larger optoelectronic network.

- In the Discussion section of the manuscript, we rephrased the following sentence:
“However, the ID₂N₂s could be incorporated in a more complex optoelectronic scheme, where the response of the detector module would play the role of the first nonlinear hidden layer of the network[Zhou et al., *Nat. Photonics* 15, 367–373 (2021)].”
With this:
“However, the ID₂N₂s could be incorporated in a more complex optoelectronic scheme, where the response of the detector module would play the role of the first nonlinear hidden layer of the network[Sitzmann, V. et al., *ACM Trans. Graph.* 37, 114 (2018), Martel, et al., *IEEE Trans. Pattern Anal. Mach. Intell.* 42, 1642–1653 (2020), Wetzstein, G. et al., *Nature* 588, 39–47 (2020)].”

Also, the authors pointed out the number of pixels as the main reason for not demonstrating the functionality of the CMOS prototype, but this is not fully convincing to me because I guess they can create another dataset with small Zernike images. These still make me hesitate to recommend this paper for publication in Nature Comms.

Response:

The reviewer suggests that he cannot recommend this manuscript for publication unless the (limited) functionality of the CMOS prototype is demonstrated. For this purpose, we added the following sections to the manuscript and supplementary information, showing the limited functionality of the CMOS prototype as well as describing the fabrication process and materials used in detail.

Changes to the manuscript:

- We added the following sentence in the *Four-layer ID₂N₂ printed on CMOS* section of the manuscript:
“The details of the fabrication process and the characterization results for a prototype device with limited functionality are presented in **Figure S11**, **Figure S12** and in the *CMOS prototype* section of the Supporting Information.”
- We replace the following sentence in the *Discussion* section of the manuscript:
“The size of the diffractive elements presented here (30×30 μm²) corresponds to 27×27 pixels on a Sony IMX219PQ CMOS image sensor with pixel size 1.12×1.12 μm², these images would be too low in resolution to appreciate the functionality of the optical network. The minimum number of pixels on the CMOS sensor required to evaluate the functionality of the network would be 75×75 pixels, which is the resolution of the training datasets. This would require diffractive elements sized 84×84 μm², which is not only challenging to achieve with the TPN method in a laboratory environment due to long fabrication times but would also require to consider nonuniformities in CMOS detector response during a post-processing step.”
With the following:
“In our current method for designing the ID₂N₂, all the layers of the network (i.e. input layer, diffractive layers, output layer) have the same number of pixels and the same pixel pitch. In our case the pixel size and pitch were chosen to maximize the performance of the DN₂ for the chosen operative wavelength (785 nm) without considering the pixel size/pitch of the CMOS sensor.

For an optimised design that consider the features of the CMOS sensor, several solutions can be considered. For example, the design of the diffractive layer could be adjusted to match the CMOS pixel size. This would result in diffractive neurons with large diameter compared with the operative wavelength, and consequently a low diffraction efficiency and an increased distance required between layers to form fully connected layers. Alternatively, the method for training the DN_2 could be extended to consider layers with different numbers of pixels, for example through changing the sampling of the fields after propagation between the respective layers.”

Changes to the supplementary information:

- We added Figure S11 and Figure S12 reporting the details of the fabrication on CMOS and the characterization results for a prototype device with limited functionality.

Figure S11. CMOS prototype fabrication and characterization. **a)** Raspberry Pi Camera Module with Sony IMX219 NoIR CMOS image sensor. **b)** The image sensor is detached from the camera module and the lens and lens support are removed in preparation for the printing process. **c)** After the printing process, the image sensor is mounted on the camera module and positioned in front of the 10X objective. **d)** ID_2N_2 (bottom view) imaged through the Sony IMX219 NoIR CMOS image sensor illuminated under an angle to the surface normal of the sensor. **e)** Optical microscope image (reflection mode) of the ID_2N_2 (top view).

Figure S12. CMOS prototype results. **a)** Original pupil phases (with RMS absolute magnitude of 0.6π radians) imposed to the point spread functions (PSFs), **(b)** experimentally detected PSF intensities and **(c)** ID₂N₂ outputs for for Z₀, Z₁, Z₂, Z₄ and -Z₄ Zernike polynomials experimentally detected by the Sony IMX219 NoIR CMOS image sensor. The ID₂N₂ was printed directly on the CMOS sensor and the images of the experimentally detected PSF and wavefront predictions consist of 27×27 pixels ($30 \times 30 \mu\text{m}^2$).

- We added the *CMOS prototype* section discussing the printing on CMOS process in detail and the results.

‘CMOS prototype

Starting from a commercial Raspberry Pi Camera Module, as shown in **Figure S11a**, we first mechanically remove the lens and lens housing using metal tweezers (**Figure S11a** and **S11b**), before detaching the CMOS submodule (**Figure S11b** top right), which carries a Sony IMX219 NoIR CMOS sensor, and cleaning the sensor surface with isopropanol. During this step one must pay close attention not to damage the wire bonds between the CMOS sensor and the PCB (printed circuit board) of the submodule. To print the ID₂N₂ the CMOS submodule was mounted on a microscope slide, commercial IP-S photoresist was deposited directly on the CMOS sensor, and photopolymerised using the TPN method in a dip in approach (as described in the *Vectorial two-photon nanolithography* section of this Supporting Information). The fabricated ID₂N₂ as imaged through the CMOS sensor illuminated under an angle to the surface normal of the sensor is shown **Figure S11d**, where one can clearly identify where the ID₂N₂ was printed on the CMOS sensor as square blocks highlighted in yellow color, while the shadow thrown on the sensor by the diffractive network under angled illumination is showing that it is indeed is a 3-Dimensional structure printed on the CMOS sensor. **Figure S11e** shows a top-view of the printed diffractive network on the CMOS sensor.

To characterize the ID₂N₂ module, we re-attached the sensor submodule with the diffractive elements printed on the CMOS sensor to the camera base module and aligned the camera module in front of the 10X objective using a sample holder as shown

in **Figure S11c**. The results of the characterization of the ID₂N module prototype on the CMOS sensor for Z₀, Z₁, Z₂, Z₄ and -Z₄ Zernike polynomials are reported in **Figure S12**, where the respective aberration functions are shown in **Figure S12a**. Although the resolution is notably limited on the Sony IMX219 NoIR CMOS sensor, the PSFs recorded on this CMOS sensor shown in **Figure S12b** are qualitatively in good agreement with the recorded PSFs as shown in **Figure 4** and **Figure 5** of the main text. The output of the ID₂N₂ module shown in **Figure S12c** qualitatively allows for an estimation of the applied Zernike pupil phase function, including the sign of the defocus.

While the results presented in this section show that it is indeed feasible to fabricate a functioning prototype of our ID₂N₂ module integrated on a CMOS sensor, the challenges in the fabrication method as outlined above are manifold. For one, the mechanical removal of the lens and lens housing from the sensor module prior to fabrication is likely (~50%) to damage the sensor wire bonds, rendering the sensor useless. Although this would be not an issue in a more advanced fabrication scheme, where one starts off a 'bare' CMOS sensor, it makes the overall fabrication process described here unreliable. Further, with the TPN printing setup as described in the *Nanoprinting* section of the manuscript being configured for imaging of the substrate in transmission mode, identifying the surface of an opaque substrate is difficult, leading to potentially large errors (+/- 15 μm) in the distance between last diffractive element and detector plane, resulting in reduced performance of the ID₂N₂ module prototypes compared to the results of the benchtop demonstration shown in the main text.

Overall, this section shows the feasibility of fabrication of the ID₂N₂ prototype integrated on a CMOS sensor and its – while limited – functionality in directly determining pupil phase distortions, including the sign of the defocus aberration, from an incident point spread function.'

Reviewer #3 (Remarks to the Author):

I appreciate the thoughtful responses by the authors, and the manuscript has been significantly improved in clarity. In principle, I support the publication of the work in Nature communications.

Response:

We are glad about the Reviewer's positive comments on our work, as well as the recommendation for publication. We also thank the Reviewer for their constructive feedback. We hope to be able to address the Reviewer's concerns in the revised manuscript as described in detail below.

While most of my concerns have been satisfyingly addressed, there are several remaining details I wish the authors can revise on.

1. The authors have pointed out that, according to Ref. 45 (Leutenegger, M., et al., Fast focus field calculations), the transformation from the complex-value E field of the PSF (E_{PSF}) to the complex-value E field at the aperture (E_{Aperture}) is a linear one.

However, at the output of the diffractive neural network, what is measured is the intensity $|E_{\text{Aperture}}|^2$ instead of the complex-value E field, and what is intended to be measured at the output of the diffractive neural network is the phase of the complex-valued E-field $\arg(E_{\text{Aperture}})$. First, these two quantities are nominally different, and I assume only under some conditions they are approximately equal. Second, it is not immediately obvious the linear relation between E_{PSF} and E_{Aperture} translates to the linear relation between E_{PSF} and $|E_{\text{Aperture}}|^2$ (or $\arg(E_{\text{Aperture}})$, if they are proven to be equal under some conditions). I hope the authors can clarify this point more.

Response:

We thank the Reviewer for the question that allows us to better explain our work.

We would like to clarify that the electric field at the aperture is not measured. What we measure is the square magnitude of a complex electric field that was linearly transformed (through an objective, that forms a PSF and then further through a DN_2 that forms a distinct complex field pattern at its output plane) such that $|E_{\text{output}}(x_i, y_j)|^2 \sim \arg(E_{\text{aperture}}(x_i, y_j))$, i.e. the intensity distribution measured at the output plane of the diffractive neural network is a linear representation of the phase at the pupil plane, the aperture phase is not directly measured. In this way we *directly* obtain the pupil phase in the sense that the phase distribution is shown immediately as a distribution over (x, y) in the output plane of the DN_2 , rather than being represented as coefficients of Zernike polynomials as done in *indirect* pupil phase retrieval schemes.

To clarify this concept:

- We rephrased the following sentence at the end of *Experimental performance* section of the manuscript:
“For this purpose, we assign values between -0.6π and 0.6π to the normalised output intensity pattern reported in **Figure 4** and **Figure S9**, where the lowest intensity refers to -0.6π and the highest intensity to $+0.6 \pi$ and the measured intensity representing the phase in π radians scales linearly with intensity.”

- We rephrased the first paragraph of the Results section of the manuscript:
 “The DN₂ receives in input a complex field that was linearly transformed through an objective that forms a PSF, which is a linear expression of the pupil phase distortion in form of a single Zernike polynomial or a superposition of multiple Zernike polynomials. Retaining phase and amplitude information of the incoming light, the DN₂ can scatter and modulate each of a multitude of aberrated PSFs, mapping them into a specific output field. The intensity distribution of the output field is a linear representation of the original pupil phase in magnitude and sign. Through this optical inference process, the pupil phase is directly obtained in the sense that the phase distribution is shown immediately as a distribution over (x, y) on the output plane of the DN₂, rather than being represented as coefficients of Zernike polynomials, as done in *indirect* phase retrieval schemes.”

2. In results section, the authors make the following statement: “While the nonlinear activation in the optical characterization of the printed DN₂s is implemented through the photoelectric conversion of the field incident on the CCD sensor³⁹, this behavior is approximated as a Rectified Linear Unit (ReLU) function of optimized shape (see Figure S1 for optimization) for purposes of in-silico training and numerical characterization of the DN₂s. The choice to use the ReLU function to approximate the nonlinearity of the detector is motivated by the response curve for a light sensitive detector module consisting of photoelectric conversion and electronic readout circuit⁴⁶ (see Supplementary Information).” I suggest authors making an explicit statement here like “*While the diffractive neural network performs a linear operation on the light field, the nonlinear response of the CCD sensor at the output of the diffractive neural network may serve as nonlinear activation function of a hidden layer, if the readout of the sensor is used as the input to another digital or optical neural network.*” This is because the role of the nonlinear response of the CMOS sensor is really determined by the subsequent operations, whether it functions as a nonlinear activation in a hidden layer or not is decided by whether there are more layers after it. If it is the last layer, the nonlinear response of the camera only serves as a bias term, as we have discussed. Also, the ReLUlike nonlinear response is not the only way to implement nonlinear function with camera sensors, in fact, the intensity detection is a more basic one and can serve as a quadratic nonlinearity, as mentioned in several previous works (Hamerly, Ryan, et al. "Large-scale optical neural networks based on photoelectric multiplication." *Physical Review X* 9.2 (2019): 021032. Also, Spall, James, Xianxin Guo, and Alexander I. Lvovsky. "Hybrid training of optical neural networks." arXiv preprint arXiv:2203.11207 (2022).) Again, I believe the impact of this work does not hinge on whether there is a nonlinear activation implemented by the camera sensor or not, but on the application of diffractive neural network for direct phase retrieval.

Response:

We thank the reviewer for helping us clarify this point in our manuscript and agree with the changes proposed by the Reviewer. We added the following statement to the Results section of the manuscript:

“While the diffractive neural network performs a linear operation on the light field, the nonlinear response of the CCD sensor at the output of the DN₂ may serve as nonlinear activation function of a hidden layer, if the readout of the sensor is used as input to another optical or digital neural network.”

REVIEWERS' COMMENTS

Reviewer #2 (Remarks to the Author):

All the issues raised in the previous round have been adequately addressed with new experiments in this revision - recommend acceptance!

Reviewer #3 (Remarks to the Author):

I would like to thank the authors for addressing my questions, and I support the publication of the manuscript.

The following response to the Reviewers comments for our manuscript “Direct retrieval of Zernike-based pupil functions using integrated diffractive deep neural networks” (NCOMMS-22-23143) is color-coded as follows:

- the original comments of the Reviewers are written in black
 - our responses to the Reviewers comments are written in blue
 -
-

REVIEWERS' COMMENTS

Reviewer #2 (Remarks to the Author):

All the issues raised in the previous round have been adequately addressed with new experiments in this revision - recommend acceptance!

Response

We thank the reviewer for the time dedicated to review our work, the constructive comments as well as the recommendation for publication. We believe that her/his feedback helped us to significantly improve our manuscript.

Reviewer #3 (Remarks to the Author):

I would like to thank the authors for addressing my questions, and I support the publication of the manuscript.

Response

We are glad about the reviewer’s recommendation for publication. We also thank the reviewer for their constructive feedback, which helped us improve our manuscript significantly.